# Mitochondrial fission during mitophagy requires both inner and outer mitofissins

Kentaro Furukawa [1✉], Tatsuro Maruyama [2], Yuji Sakai [3], Shun-ichi Yamashita [1], Keiichi Inoue [1], Tomoyuki Fukuda [4], Nobuo N Noda [2,5] & Tomotake Kanki [1✉]

## Abstract

**Mitophagy maintains mitochondrial homeostasis through the selective degradation of damaged or excess mitochondria. Recently, we identified mitofissin/Atg44, a mitochondrial inter-membrane space-resident fission factor, which directly acts on lipid membranes and drives mitochondrial fission required for mito-phagy in yeast. However, it remains unclear whether mitofissin is sufficient for mitophagy-associated mitochondrial fission and whether other factors act from outside mitochondria. Here, we identify a mitochondrial outer membrane-resident mitofissin-like microprotein required for mitophagy, and we name it mitofissin 2/Mfi2 based on the following results. Overexpression of an N-terminal Atg44-like region of Mfi2 induces mitochondrial frag-mentation and partially restores mitophagy in *atg44Δ* cells. Mfi2 binds to lipid membranes and mediates membrane fission in a cardiolipin-dependent manner in vitro, demonstrating its intrinsic mitofissin activity. Coarse-grained molecular dynamics simulations further support the stable interaction of Mfi2 with cardiolipin-containing bilayers. Genetic analyses reveal that Mfi2 and the dynamin-related protein Dnm1 independently facilitate mitochon-drial fission during mitophagy. Thus, Atg44 and Mfi2, two mito-fissins with distinct localizations, are required for mitophagy-associated mitochondrial fission.**

**Keywords** Atg44; Mfi2; Mitochondrial Fission; Mitofissin; Mitophagy
**Subject Categories** Autophagy & Cell Death; Membranes & Trafficking; Organelles

## Introduction

Autophagy is a catabolic process that mediates vacuolar/lysosomal degradation and recycling of various cytoplasmic components, including organelles. Upon induction of autophagy, double-membranous structures called isolation membranes or phagophores emerge in the cytosol, expand, and engulf cytoplasmic proteins and organelles to form autophagosomes. The autophago-some fuses with vacuoles in yeast and plants or lysosomes in mammals, leading to degradation of its sequestered materials by vacuolar/lysosomal hydrolases (Nakatogawa, 2020). Mitophagy is a type of autophagy that selectively degrades damaged or excess mitochondria and contributes to mitochondrial homeostasis (Pickles et al, 2018; Onishi et al, 2021; Uoselis et al, 2023). Autophagy and mitophagy share several molecular processes, but the latter requires receptor-mediated recognition of particular mitochondrial regions as cargoes. Various mitochondrial outer membrane-resident receptors have been reported from yeast (Kanki et al, 2009; Okamoto et al, 2009; Fukuda et al, 2020) to mammals (Novak et al, 2010; Hanna et al, 2012; Liu et al, 2012; Murakawa et al, 2015; Yoo et al, 2020). The size of the mitochondrial cargo is also critical for mitophagy. Because mitochondria are typically larger than the autophagosome, mitochondria need to be fragmented to allow their engulfment within the autophagosome (Ashrafi and Schwarz, 2013; Liesa and Shirihai, 2013; Sebastián et al, 2017; Shirihai et al, 2015). However, the known mitochondrial fission factors, dynamin-related proteins Dnm1 in yeast and Drp1 in mammals, have been shown to be dispensable for mitophagy (Mendl et al, 2011; Yamashita et al, 2016; Burman et al, 2017), suggesting the involvement of an alternative fission mechanism in this process.

Recently, we reported that the mitochondrial intermembrane space protein Atg44 (also known as Mdi1/Mco8) drives mitochon-drial fission during mitophagy in the yeast *Saccharomyces cerevisiae*, and that Atg44 directly binds to lipid membranes and induces membrane fragility to facilitate membrane fission (Fukuda et al, 2023). Therefore, we named this protein "mitofissin" (*mito*chondrial *fiss*ion prot*in*). Unlike the mammalian "mitofusin" family, which is involved in mitochondrial fusion, "mitofissin" refers to a family of proteins, like Atg44, that play a role in mitochondrial fission. In addition, Atg44 is required for the completion of Dnm1-mediated mitochondrial fission under homeostatic conditions (Connor et al, 2023; Furukawa et al, 2024), suggesting coordinated actions of these two fission factors from the inside and the outside of mitochondria, respectively. However, these findings do not fully explain why Dnm1 is

[1]Department of Cellular Physiology, Graduate School of Medical Sciences, Kyushu University, Fukuoka 812-8582, Japan. [2]Institute of Microbial Chemistry (BIKAKEN), Shinagawa-ku, Tokyo 141-0021, Japan. [3]School of Science/Graduate School of Nanobioscience, Yokohama City University, Yokohama, Kanagawa 236-0027, Japan. [4]Department of Cellular Physiology, Niigata University Graduate School of Medical and Dental Sciences, Niigata 951-8510, Japan. [5]Institute for Genetic Medicine, Hokkaido University, Sapporo, Hokkaido 060-0815, Japan. ✉E-mail: furukawa.kentaro.828@m.kyushu-u.ac.jp; kanki.tomotake.114@m.kyushu-u.ac.jp

dispensable for mitophagy. Moreover, it remains unclear whether Atg44 is sufficient for mitophagy-associated mitochondrial fission.

In this study, we identified a mitochondrial outer membrane-resident microprotein, Mco12, that is partially required for mitophagy. In vivo and in vitro analyses revealed that Mco12 possesses mitofissin activity, and we therefore renamed this protein "*mito*fissin 2" (Mfi2). In vitro assays showed that Mfi2 exhibits lipid membrane-binding and -fission activity preferentially on cardiolipin-containing membranes, and coarse-grained molecular dynamics simulations further supported its stable association with such membranes. Furthermore, we found that Mfi2 and Dnm1 facilitate mitochondrial fission in parallel from outside the mitochondria during mitophagy, providing an explanation for why disruption of Dnm1 alone does not prevent mitophagy. We propose that inner and outer mitofissins, Atg44 and Mfi2, together with Dnm1, coordinate mitophagy-associated mitochondrial fission from inside and outside the mitochondria.

## Results and discussion

### Identification of Mco12/Mfi2 as an Atg44-like microprotein

Since mitofissin/Atg44 was one of the previously unexplored microproteins, we expected that other microproteins might also play a role in mitophagy. Therefore, we systematically examined mitophagy in yeast cells lacking mitochondrial micro- or small proteins identified in previous mitochondrial proteomic analyses (Morgenstern et al, 2017) (Table EV1). In yeast, mitophagy can be induced either by nitrogen starvation (SD-N medium) or by continuous culture in a non-fermentable medium (YPL medium) to stationary phase, and can be monitored by assessing the vacuolar processing of chimeric mitochondrial proteins, such as Om45-GFP and Idh1-GFP, resulting in the release of free GFP (Kanki and Klionsky, 2008; Kanki et al, 2009). We found that among the mutant cells analyzed, *mco12Δ* (*mfi2Δ*) cells exhibited slightly decreased mitophagy upon nitrogen starvation compared with wild-type (WT) cells (Fig. EV1A). We confirmed this slight decrease by a time course assay (Fig. 1A, *mfi2Δ*) and observed a similar decrease in mitophagy induced during the stationary phase (Fig. 1B, *mfi2Δ*). Thus, we concluded that Mco12 is partially required for mitophagy. Next, we analyzed mitochondrial morphology by fluorescence microscopy in cells lacking mitochondrial micro- or small proteins in YPL medium. Unlike *atg44Δ* cells with enlarged mitochondria, the other mutant cells showed fragmented mitochondria similar to those in WT cells (Fig. EV1B).

We found two similarities between Atg44 and Mco12. First, these proteins share a low but significant sequence homology (Fig. EV1C). Second, AlphaFold prediction (Jumper et al, 2021) revealed a structural similarity between the full length of Atg44 and the N-terminal two-thirds of Mco12 (Fig. EV1D). Notably, Mco12 possesses a C-terminal disordered region that is absent in Atg44 (Fig. EV1D), suggesting that Mco12 has distinct features from Atg44. CLIME analysis (Li et al, 2014) indicated that Mco12 is less conserved among fungal species than Atg44 (Fig. EV2). On the basis of its similarity to Atg44 and the functions described below, we renamed Mco12 as "*mito*fissin 2" (Mfi2).

### Mfi2 is a mitochondrial outer membrane protein

To characterize Mfi2, we first generated antibodies to detect endogenous Mfi2 (Fig. 1C). We found that, unlike the typical mitochondrial proteins Cox2 and Idh1, Mfi2 protein levels did not differ significantly in fermentation (YPD) and respiration (YPL) media (Fig. 1C). Next, we examined the subcellular localization of Mfi2 by biochemical fractionation. As expected, Mfi2 was detected in the mitochondria-enriched fraction along with Cox2, but not in the cytosol-enriched fraction containing Pgk1 (Fig. 1D). We then analyzed the intramitochondrial localization of Mfi2 using a proteinase K (ProK) protection assay. This assay evaluates whether the mitochondrial outer or inner membrane protects proteins from proteolytic cleavage, thereby indicating their submitochondrial localization. Degradation of Mic60 (intermembrane space, IMS) and Atp2 (mitochondrial matrix) required ProK treatment in combination with hypo-osmotic swelling and/or Triton X-100 lysis, whereas ProK treatment alone caused degradation of Mfi2 and the outer membrane protein Atg33 (Fig. 1E). We also performed a sodium carbonate extraction assay, which can reveal whether a protein is embedded within the lipid bilayer (integral) or associated with the membrane surface (peripheral). We treated the mitochondrial fraction with sodium carbonate and separated the membrane and supernatant fractions by ultracentrifugation. Mfi2 was detected in the pellet fraction along with the integral membrane protein Cox2, but not in the supernatant fraction containing the IMS protein Atg44 and the peripheral membrane protein Atp2 (Fig. 1F). This result suggests that Mfi2 is strongly associated with the mitochondrial membrane. Topology prediction identified only a weakly hydrophobic N-terminal region of Mfi2 (Fig. EV1E), suggesting that Mfi2 associates with the outer membrane in a monotopic rather than classical transmembrane manner. Taken together, we conclude that Mfi2 is a mitochondrial outer membrane-resident protein.

### Mfi2 has the potential to promote mitochondrial fission in vivo

Next, we analyzed the significance of the N-terminal Atg44-like and C-terminal disordered regions of Mfi2. We expressed GFP-fused full-length, C- and N-terminal truncated Mfi2 proteins and a mutant lacking the first α-helix (Δα1) (Fig. 2A) and verified their expression (Fig. 2B). As shown in Fig. 2C, GFP-Mfi2 and the C-terminal truncated GFP-Mfi2(N66) were co-localized with the mitochondrial marker Om14-RFP, whereas the N-terminal truncated GFP-Mfi2(C33) was observed throughout the cytosol. GFP-Mfi2(Δα1) showed no detectable mitochondrial localization and instead mislocalized to the plasma membrane. These results indicate that the Atg44-like region, particularly the first α-helix, is required for mitochondrial localization of Mfi2, whereas the C-terminal disordered region is dispensable.

We hypothesized that Mfi2(N66) functions similarly to Atg44, and therefore examined whether Mfi2(N66) can rescue the mitophagy defect of *atg44Δ* cells. Overexpression of full-length Mfi2 did not rescue mitophagy of *atg44Δ* cells, whereas overexpression of Mfi2(N66) slightly rescued the mitophagy defect (Fig. 2D). In addition, overexpression of Mfi2(N66), but not full-length Mfi2, induced mitochondrial fragmentation in *atg44Δ* cells, which typically exhibit enlarged mitochondria (Fukuda et al, 2023; Connor et al, 2023; Furukawa et al, 2024) (Fig. 2E). Immunoblot analyses detected

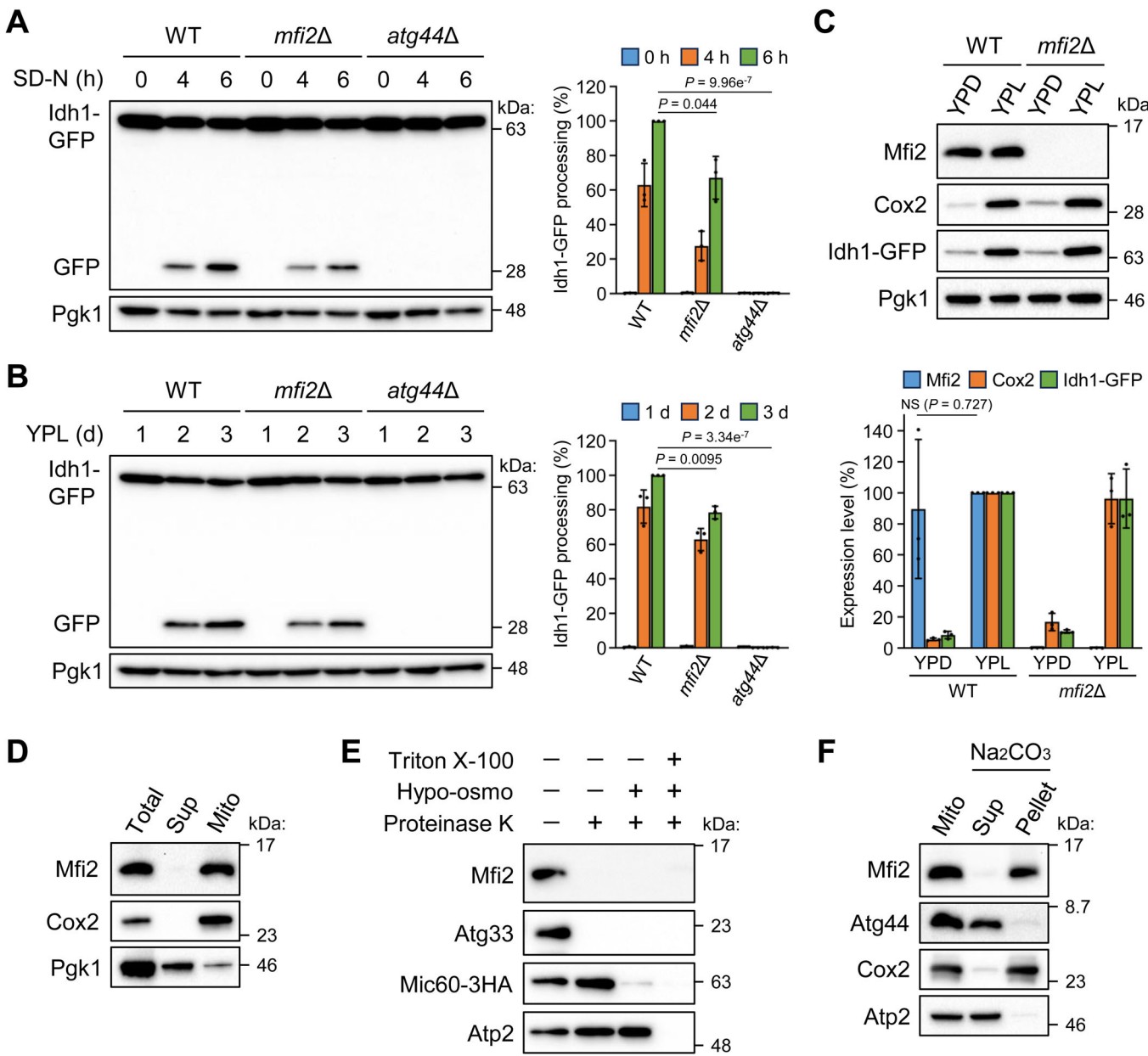

**Figure 1. Mfi2 is a mitochondrial outer membrane-resident protein partially required for mitophagy.**

(A, B) The indicated cells were cultured in YPL until mid-log phase and shifted to synthetic minimal medium lacking nitrogen (SD-N) (A) or continuously cultured in YPL (B). Cells were collected at the indicated time points, and Idh1-GFP processing was monitored by immunoblotting. Pgk1 was detected as a loading control. The value of WT at 6-h (A) or 3-d (B) time point was set to 100%. The quantification results are shown as mean ± SD ($n = 3$ biological replicates). Welch's $t$ test; exact $P$ values are shown in the graphs. (C) WT and $mfi2\Delta$ cells were cultured in YPD medium until early-log phase and in YPL until mid-log phase. Endogenous expression levels of Mfi2 were analyzed by immunoblotting. Cox2 and Idh1-GFP were detected as mitochondrial marker proteins. The value of WT (YPL) was set to 100%. The quantification results are shown as mean ± SD ($n = 3$ biological replicates). Welch's $t$ test; exact $P$ value is shown in the graph. NS, not significant. (D) Subcellular fractionation was conducted using WT cells. The total cell homogenate was fractionated by centrifugation to obtain a mitochondria-enriched pellet and supernatant. Cox2 and Pgk1 were detected as markers of the mitochondria and cytosol, respectively. (E) Isolated mitochondria from cells expressing Mic60-3HA were treated with (+) or without (−) proteinase K under different conditions. Hypo-osmotic swelling resulted in rupture of the mitochondrial outer membrane, and the detergent Triton X-100 lysed mitochondria. Atg33, Mic60-3HA, and Atp2 were detected as markers of the mitochondrial outer membrane, intermembrane space, and matrix, respectively. (F) Isolated mitochondria were treated with sodium carbonate and separated into the soluble supernatant and membrane pellet by ultracentrifugation. Cox2 and Atp2 were detected as markers of integral and peripheral membrane proteins, respectively. Source data are available online for this figure.

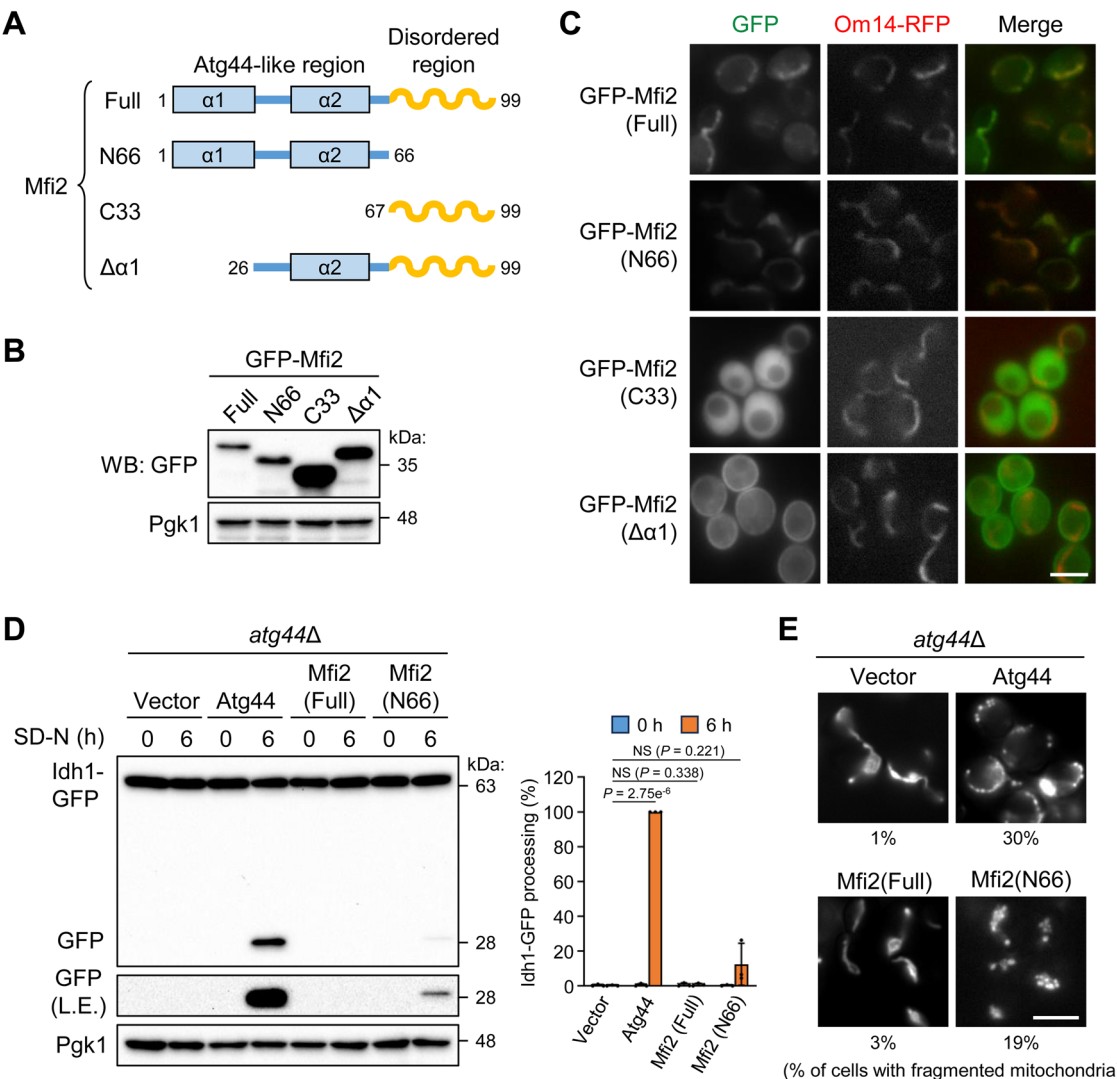

**Figure 2. Mfi2 has the potential to promote mitochondrial fission in vivo.**

(A) Schematic representation of the N-terminal Atg44-like region (blue) and the C-terminal disordered region (orange) of Mfi2, and truncated forms. (B) Cells expressing GFP-fused Mfi2 derivatives under the *ADH1* promoter were cultured in YPD until early-log phase, and their protein expression was verified by immunoblotting. (C) Cells expressing GFP-fused Mfi2 derivatives were cultured in YPD until early-log phase, and their localization was analyzed by fluorescence microscopy. Om14-RFP was detected as a mitochondrial marker. Scale bar, 4 μm. (D) *atg44Δ* cells expressing the indicated proteins under the *TDH3* promoter were cultured in SML until mid-log phase and shifted to SD-N. Cells were collected at the indicated time points, and Idh1-GFP processing was monitored by immunoblotting. The value of Atg44 (6 h) was set to 100%. The quantification results are shown as mean ± SD (*n* = 3 biological replicates). Welch's *t* test; exact *P* values are shown in the graph. NS, not significant. (E) WT cells and *atg44Δ* cells expressing the indicated proteins were cultured in SMD until mid-log phase, and their mitochondrial morphology was analyzed by fluorescence microscopy. The percentage of cells with fragmented mitochondria is shown. The total number of cells analyzed from four independent experiments was as follows: Vector, *n* = 405; Atg44, *n* = 347; Mfi2(Full), *n* = 357; Mfi2(N66), *n* = 319. Scale bar, 4 μm. Source data are available online for this figure.

exogenously expressed full-length Mfi2 and Mfi2(N66) at much higher levels than endogenous Mfi2 (Fig. EV3), indicating that the Atg44-like activity of Mfi2(N66) is manifested only under overexpression conditions. Taken together, these results indicate that Mfi2 has the potential to promote mitochondrial fission in vivo.

## Mfi2 possesses lipid membrane-binding and -fission activity in vitro

To characterize Mfi2 in vitro, we prepared recombinant maltose-binding protein (MBP)-fused Mfi2 and MBP-Atg44 as a control (Fig. EV4A). Size-

exclusion chromatography coupled with multi-angle light scattering (SEC-MALS) analysis estimated the molar masses of soluble MBP-Atg44 and MBP-Mfi2 as 1.16 and 0.99 MDa, respectively (Fig. EV4B). Their calculated molecular masses are 50 and 54 kDa, respectively, indicating that these recombinant proteins are highly oligomerized in solution. Using the recombinant proteins, we first examined whether Mfi2 binds to lipid membranes. Both Alexa Fluor 488 (AF488)-labeled MBP-Atg44 and AF647-MBP-Mfi2 accumulated on a giant unilamellar vesicle (GUV) derived from a lipid film with a phospholipid composition similar to that of the inner mitochondrial membrane (Fig. 3A), indicating that Mfi2 also possesses lipid membrane-binding activity. Importantly, the membrane

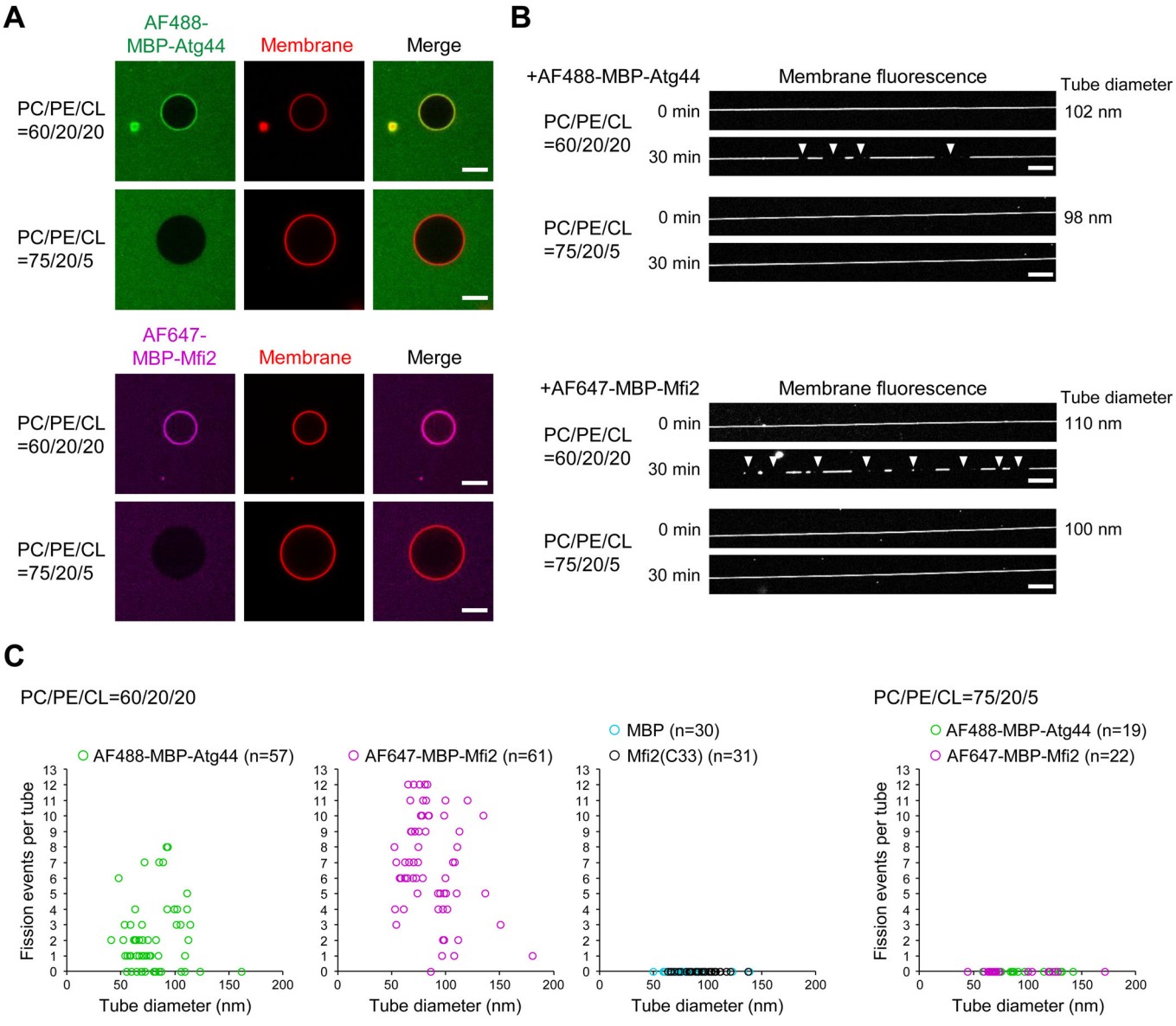

**Figure 3. Mfi2 possesses membrane-binding and -fission activity in vitro.**

(A) Membrane binding of MBP-Atg44 and MBP-Mfi2. Membrane binding was examined by confocal laser scanning microscopy using fluorescently labeled proteins and GUVs labeled with liss Rhod PE. Scale bars, 10 μm. (B) Membrane fission by MBP-Atg44 and MBP-Mfi2. Fission of lipid nanotubes was observed by confocal laser scanning microscopy using fluorescently labeled proteins and lipid nanotubes labeled with liss Rhod PE. Positions of fission are marked with arrowheads. Scale bars, 10 μm. (C) Quantification of tube fission assays. For each lipid nanotube, the number of fission events per tube within the region of interest (ROI) was plotted against tube diameter. MBP and Mfi2(C33) served as negative controls. Source data are available online for this figure.

binding of both Mfi2 and Atg44 was severely impaired when cardiolipin (CL) was reduced or absent in the phospholipids, suggesting that their membrane binding in vitro requires CL (Figs. 3A and EV4C).

We then prepared lipid nanotubes fluorescently labeled with liss Rhod PE (Fukuda et al, 2023; Maruyama and Noda, 2023) and used them to analyze the membrane fission activity of Mfi2. Upon application of AF647-MBP-Mfi2 as well as AF488-MBP-Atg44, most of the nanotubes underwent fission at multiple sites. This fission event was strongly dependent on the presence of a high concentration of CL (Fig. 3B). Quantitative analyses further confirmed that both Atg44 and Mfi2 induced frequent and reproducible fission events under high CL-containing conditions (Fig. 3C). As expected, MBP alone and the mitochondrial targeting-deficient mutant Mfi2(C33) did not induce fission events at all, thereby serving as negative controls (Fig. 3C). These data indicate that Mfi2, like Atg44, possesses membrane fission activity in vitro.

## CGMD simulations of Mfi2–lipid bilayer interactions

The in vitro assays described above revealed that Mfi2 requires CL in the phospholipid composition for membrane binding and nanotube fission (Fig. 3). CL is normally enriched in the

mitochondrial inner membrane, but has been reported to translocate to the outer membrane under mitophagy-inducing conditions (Chu et al, 2013), providing a physiological context for the CL-dependent membrane association of the outer membrane protein Mfi2. To further investigate the molecular basis of this CL dependence, we performed coarse-grained molecular dynamics (CGMD) simulations of Mfi2–lipid bilayer interactions with or without CL. In simulations with CL-containing bilayers (PC:PE:CL = 60:20:20), full-length Mfi2 was initially placed 3 nm above the bilayer and diffused randomly. Upon contact, Mfi2 associated with the bilayer and remained bound throughout the 1-µs simulation (Fig. 4A; Movie EV1). In contrast, in the absence of CL (PC:PE:CL = 80:20:0), Mfi2 repeatedly approached the bilayer but did not establish a stable association with it (Fig. 4B; Movie EV2). Similar CL-dependent behavior was observed for the C-terminally truncated mutant Mfi2(N66). Mfi2(N66) also associated with CL-containing bilayers (Fig. 4C; Movie EV3), but did not stably interact with bilayers lacking CL (Fig. 4D; Movie EV4). These results indicate that CL is critical for stable bilayer association of Mfi2, and that the N-terminal region alone is sufficient for this CL-dependent interaction.

To further validate the CL-dependent membrane targeting of Mfi2, we analyzed GFP-Mfi2 localization in crd1Δ cells, in which CL biosynthesis is blocked. Although mitochondrial localization of GFP-Mfi2 was observed in crd1Δ cells as in WT cells, cytoplasmic GFP puncta were frequently detected in the mutant (Fig. 4E), suggesting impaired mitochondrial targeting of Mfi2. These observations indicate that efficient mitochondrial targeting of Mfi2 requires CL in vivo, and are consistent with the in vitro assays and CGMD results.

## Mfi2 and Dnm1 independently contribute to mitophagy-associated mitochondrial fission

Dnm1 is the first identified mitochondrial fission factor (Otsuga et al, 1998; Bleazard et al, 1999), but it is dispensable for mitophagy (Mendl et al, 2011; Yamashita et al, 2016; Fukuda et al, 2023). Mfi2 is also not essential for mitophagy (Fig. 1A,B). These features led us to hypothesize that Mfi2 and Dnm1 redundantly contribute to mitochondrial fission during mitophagy. Therefore, we examined mitophagy in mfi2Δ dnm1Δ double mutant cells. As shown in Fig. 5A, mfi2Δ and dnm1Δ single mutant cells exhibited slightly decreased mitophagy compared with WT cells, whereas the mfi2Δ dnm1Δ double mutant cells displayed a marked decrease in mitophagy. These results suggest that Mfi2 and Dnm1 independently contribute to mitophagy.

Taking advantage of the impaired mitophagy in mfi2Δ dnm1Δ cells, we examined two Mfi2 orthologs from Ashbya gossypii (Ag) and Debaryomyces hansenii (Dh) (Figs. EV2 and EV5A). Despite structural differences in the C-terminal regions predicted by AlphaFold (Fig. EV5B), expression of AgMfi2 or DhMfi2 restored mitophagy in mfi2Δ dnm1Δ cells to levels comparable to those of Mfi2 (Fig. EV5C). These results suggest that the C-terminal region of Mfi2 does not play a primary role in its function in mitophagy. We also compared the functions of Mfi2(N66) and Atg44 with Mfi2. Expression of Mfi2 or Mfi2(N66) restored mitophagy in mfi2Δ dnm1Δ cells at the same level, whereas expression of Atg44 did not (Fig. 5B). These results suggest that the C-terminal region of Mfi2 is dispensable for its function and that Atg44, despite

comparable in vitro activity, cannot fulfill the role of Mfi2 acting from outside the mitochondria. Taken together with the result that full-length Mfi2 cannot restore mitophagy in atg44Δ cells (Fig. 2D), we conclude that both inner (Atg44) and outer (Mfi2) mitofissins are required for efficient mitophagy.

Next, we investigated why mitophagy is impaired in mfi2Δ dnm1Δ cells. In WT cells, mitochondrial Idh1-GFP signals were accumulated in vacuoles upon induction of mitophagy (Fig. 5C, arrows). By contrast, mfi2Δ dnm1Δ cells exhibited mitochondrial protrusions, a typical morphological phenotype of the mitochondrial fission defect during mitophagy, as observed in atg44Δ cells (Fukuda et al, 2023) (Fig. 5C, arrowheads). As in the case of the protrusions observed in atg44Δ cells, this protrusion formation in mfi2Δ dnm1Δ cells was completely suppressed by disruption of autophagy/mitophagy factors such as Atg1 (a protein kinase of the Atg1 core complex), Atg8 (a ubiquitin-like protein required for autophagosome formation), Atg11 (an adapter for selective autophagy), and Atg32 (a receptor for mitophagy) (Fig. 5D), indicating that the mitochondrial protrusion formation depends on the mitophagy process. Notably, mitochondrial protrusions were also observed in mfi2Δ dnm1Δ atg44Δ triple mutant cells without additive effects (Fig. 5D). Consistent with the autophagy-dependent protrusion formation, we additionally observed GFP-Atg8 puncta on mitochondrial protrusions in these triple mutant cells (Fig. 5E), as in atg44Δ cells described previously (Fukuda et al, 2023). These results indicate that autophagy machinery defines the mitochondrial fission sites during mitophagy, but in the absence of Atg44 or of both Mfi2 and Dnm1, these sites fail to undergo fission and remain as protrusions. Taken together, we conclude that Mfi2 and Dnm1, as well as Atg44, are required for completion of mitochondrial fission during mitophagy.

The identification of mitofissin/Atg44 as a mitochondrial fission factor essential for mitophagy raised the question of whether Atg44-mediated action from the IMS is sufficient for mitochondrial fission. In the present study, we identified Mfi2 as a mitochondrial outer membrane-resident mitofissin and found that simultaneous disruption of Mfi2 and Dnm1 resulted in a severe defect in mitophagy, suggesting that Mfi2 and Dnm1 independently contribute to mitochondrial fission during mitophagy. The mitochondrial fission defect in mfi2Δ dnm1Δ cells, even in the presence of Atg44, also suggests that Atg44 alone is not sufficient to complete mitochondrial fission. Thus, this study demonstrates that mitochondrial fission during mitophagy requires both "inner" and "outer" fission factors. The IMS mitofissin Atg44 promotes mitophagy-associated fission in coordination with the outer membrane protein mitofissin Mfi2 or the dynamin-related protein Dnm1, both acting from outside the mitochondria.

On the basis of our results and previous reports, we propose a model for mitochondrial fission during mitophagy in yeast (Fig. 5F). Upon induction of mitophagy, isolation membranes emerge on mitochondria and extend along the mitochondrial surface (Yamashita et al, 2016). This process depends on the mitochondrial loading of the autophagy machinery, which is mediated by the interaction between the mitophagy receptor Atg32 and the adapter Atg11 (Furukawa et al, 2018; Fukuda et al, 2023). The extension of isolation membranes leads to the formation of a mitochondrial bud by an as yet unidentified mechanism, apparently independently of known mitochondrial fission factors (Fig. 5D). Atg44 binds to mitochondrial membranes from the inside (Fukuda et al, 2023;

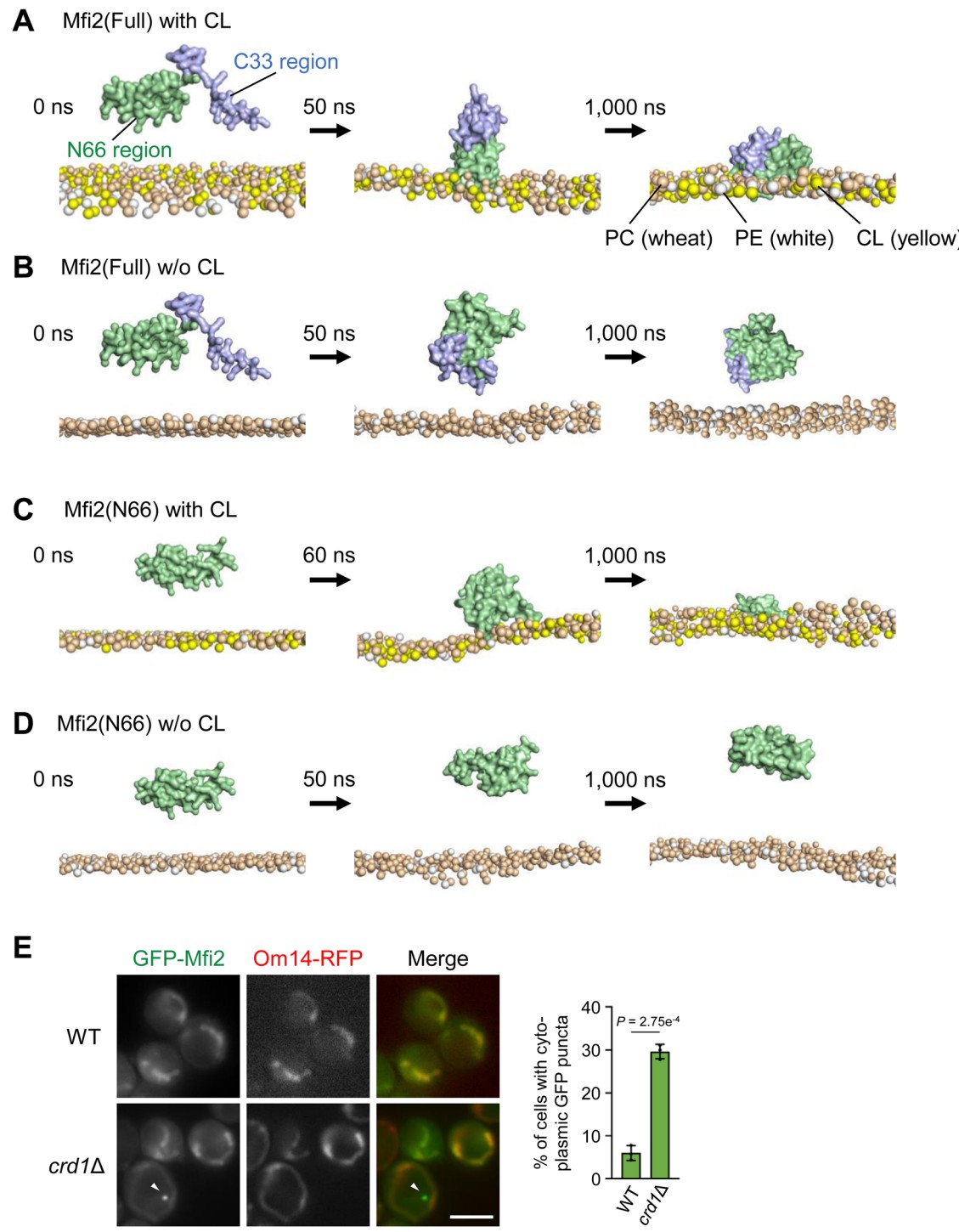

**Figure 4. CGMD simulations of Mfi2–lipid bilayer interactions.**

(**A, B**) Coarse-grained molecular dynamics simulations of full-length Mfi2 with bilayers containing CL (PC:PE:CL = 60:20:20) (**A**) or lacking CL (PC:PE:CL = 80:20:0) (**B**). Mfi2 remained stably associated only with CL-containing bilayers. (**C, D**) Similar CL-dependent behavior was observed for Mfi2(N66), which bound stably to CL-containing bilayers (**C**) but not to CL-lacking bilayers (**D**). See also Movies EV1–EV4. (**E**) WT or crd1Δ cells expressing GFP-Mfi2 under the *ADH1* promoter were cultured in YPD until early-log phase and then analyzed by fluorescence microscopy. Arrowheads indicate cytoplasmic GFP puncta in crd1Δ cells. The percentage of cells with cytoplasmic GFP puncta was quantified. The quantification results are shown as mean ± SD (*n* = 3 biological replicates). Welch's *t* test; exact *P* value is shown in the graph. The total number of cells analyzed from three independent experiments was as follows: WT, *n* = 266; crd1Δ, *n* = 270. Scale bar, 4 μm. Source data are available online for this figure.

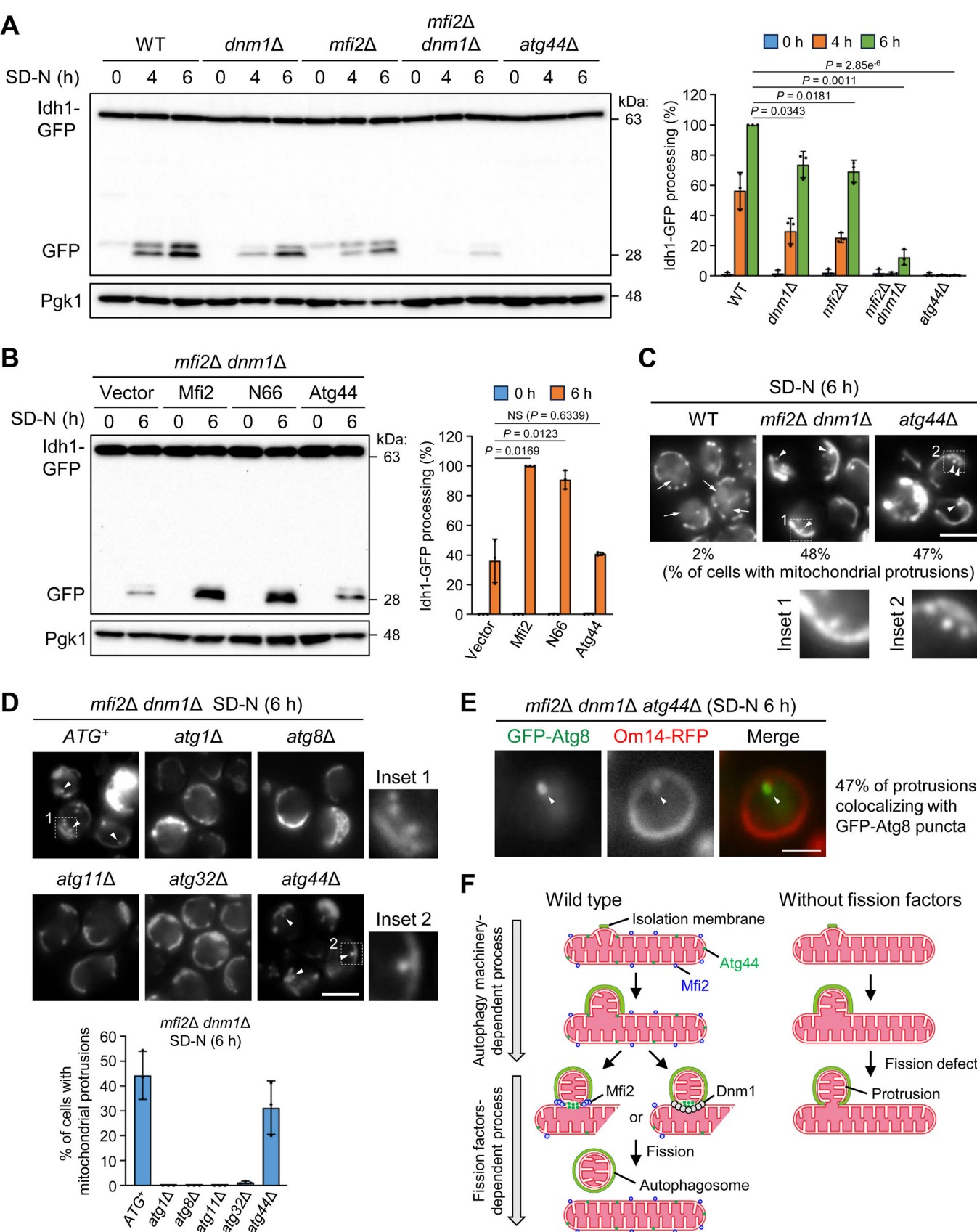

**Figure 5.  Mfi2 and Dnm1 independently contribute to mitophagy-associated mitochondrial fission.**

(A, B) The indicated cells were cultured in YPL (A) or SML (B) until mid-log phase and shifted to SD-N. Cells were collected at the indicated time points, and Idh1-GFP processing was monitored by immunoblotting. The value of WT at 6-h (A) or *dnm1Δ mfi2Δ* cells expressing Mfi2 at 6-h (B) time point was set to 100%. The quantification results are shown as mean ± SD ($n = 3$ biological replicates). Welch's *t* test; exact *P* values are shown in the graphs. NS, not significant. (C, D) The indicated cells were cultured in YPL until mid-log phase and shifted to SD-N for 6 h. The absence or presence of mitochondrial protrusions (arrowheads) was analyzed by fluorescence microscopy. Insets show higher-magnification views of representative protrusions. Vacuolar GFP accumulation is indicated by arrows (WT in C). The percentage of cells with mitochondrial protrusions was quantified. The total number of cells analyzed from three independent experiments was as follows: WT, $n = 258$; *mfi2Δ dnm1Δ*, $n = 211$; *atg44Δ*, $n = 251$ in (C). *ATG⁺*, $n = 331$; *atg1Δ*, $n = 420$; *atg8Δ*, $n = 300$; *atg11Δ*, $n = 403$; *atg32Δ*, $n = 326$; *atg44Δ*, $n = 303$ in (D). The quantification results are shown as mean ± SD in (D). Scale bars, 4 μm. (E) The *mfi2Δ dnm1Δ atg44Δ* triple mutant cells expressing Om14-RFP and GFP-Atg8 were grown in YPL and shifted to SD-N medium for 6 h and then analyzed by fluorescence microscopy. Arrowheads indicate accumulation of GFP-Atg8 on the mitochondrial protrusions, and the percentage of their colocalization was quantified. 180 cells from three independent experiments were analyzed. Scale bar, 2 μm. (F) Schematic model of mitochondrial fission upon induction of mitophagy. See the main text for details. Source data are available online for this figure.

Furukawa et al, 2024), while Mfi2 and Dnm1 bind to mitochondrial membranes from the outside. The coordinated actions of Atg44 with either Mfi2 or Dnm1 ultimately lead to mitochondrial fission, although the exact timing of this fission event remains to be determined. Finally, the mitochondrial fragment generated by this fission event is engulfed within the autophagosome. In the absence of either the internal action of Atg44 or the external action of Mfi2/Dnm1, the budded portion remains connected to the main body of the mitochondrion, resulting in protrusion formation.

Although this study advances our understanding of mitochondrial fission during mitophagy, several important questions remain unanswered. First, it is unclear how mitochondrial fission is facilitated in organisms that possess Atg44 orthologs but lack Mfi2 orthologs (e.g., *Schizosaccharomyces pombe* and most filamentous fungi) (Fig. EV2). In such organisms, it is possible that Atg44 orthologs can mediate both inner and outer mitochondrial membrane fission and/or that Dnm1 orthologs have sufficient activity for outer mitochondrial membrane fission. Comparative analysis of these orthologs across different species may provide insights that answer this question. Second, the importance of the C-terminal region of Mfi2 is unclear. The observation that the C-terminally truncated form of Mfi2 [Mfi2(N66)] is able to partially act like the IMS mitofissin Atg44 (Fig. 2D,E) suggests that the presence of the C-terminal region prevents Mfi2 import into mitochondria. Interestingly, Mfi2 orthologs such as AgMfi2 and DhMfi2 have different C-terminal structures, but they can rescue the function of Mfi2 (Fig. EV5). This result suggests that the C-terminal region of Mfi2 has an unidentified but important role, although its structure has varied during evolution. One possible role of the C-terminal region of Mfi2 is to make Mfi2 localize strictly to the outer membrane. Third, it remains unclear how Mfi2 associates with the mitochondrial outer membrane despite the low abundance of CL in this compartment. Our CGMD simulations confirmed that Mfi2 preferentially interacts with CL-containing bilayers (Fig. 4), further supporting the biochemical observation of its CL-dependent binding. However, these results seem counterintuitive given the fact that CL is predominantly contained in the mitochondrial inner membrane (~20% in the inner membrane and ~3% in the outer membrane) (Dudek, 2017). Previous reports have shown that CL externalization from the inner to the outer membrane serves as a mitophagy signal (Chu et al, 2013; Kagan et al, 2016). In addition, Drp1 has been shown to bind to CL, and this interaction is thought to be critical for its recruitment to mitochondria under stress conditions (Mahajan et al, 2021). These findings raise the possibility that Mfi2 functions in response to CL

accumulation at the outer membrane of mitochondrial fission sites. Future studies are needed to test this hypothesis. Fourth, it is not yet clear whether Mfi2 and Dnm1 function cooperatively or independently for mitochondrial fission during mitophagy. In contrast to the enlarged mitochondrial morphology observed in *atg44Δ* and *dnm1Δ* cells (Fukuda et al, 2023; Connor et al, 2023; Furukawa et al, 2024), *mfi2Δ* cells display normal mitochondrial morphology under non-mitophagy-inducing conditions (Fig. EV1B). This observation suggests that Mfi2 and Dnm1 act independently, at least under homeostatic conditions, but further studies are needed to clarify this issue.

In summary, this study identifies Mfi2 as a mitochondrial outer membrane-type mitofissin and provides a compelling explanation for why Dnm1 is dispensable for mitophagy. Further studies will advance our understanding of how mitophagy-associated mitochondrial fission is orchestrated in yeast, and shed light on mitochondrial fission in higher eukaryotes where mitofissin counterparts have not yet been identified, although they are likely to exist. Furthermore, the discovery of mitofissin-like proteins required for the fission of other organelles, such as the endoplasmic reticulum and peroxisomes, would be of great interest, and our findings may provide a useful framework for such studies.

## Methods

**Reagents and tools table**

| Reagent/resource | Reference or source | Identifier or catalog number |
|---|---|---|
| **Experimental models** | | |
| *Saccharomyces cerevisiae* | Table EV2 | N/A |
| **Recombinant DNA** | | |
| pCu416 (*CEN URA3 PCUP1*) | Labbé and Thiele, 1999 | N/A |
| pTDH416 (*CEN URA3 PTDH3*) | Fukuda et al, 2023 | N/A |
| pGEX-4T-1 | Promega | Cat# 28-9545-49 |
| pET15b | Novagen | Cat# 69661-3 |
| pCu416-ATG44 | Fukuda et al, 2023 | N/A |
| pTDH416-ATG44 | Fukuda et al, 2023 | N/A |

| Reagent/resource | Reference or source | Identifier or catalog number |
|---|---|---|
| pCu416-MFI2 | This study | N/A |
| pTDH416-MFI2 | This study | N/A |
| pCu416-MFI2(N66) | This study | N/A |
| pTDH416-MFI2(N66) | This study | N/A |
| pTDH416-MFI2-FLAG | This study | N/A |
| pTDH416-MFI2(N66)-FLAG | This study | N/A |
| pCu416-AgMFI2 | This study | N/A |
| pCu416-DhMFI2 | This study | N/A |
| **Antibodies** | | |
| Rabbit polyclonal anti-Mfi2 | This study | N/A |
| Mouse monoclonal anti-GFP (JL8) | Takara Bio | Cat# 632380, RRID:AB_10013427 |
| Mouse monoclonal anti-Pgk1 | Thermo Fisher Scientific | Cat# 459250, RRID:AB_2532235 |
| Rabbit polyclonal anti-Atp2 | Abcam | Cat# ab128743, RRID:AB_2810299 |
| Mouse monoclonal anti-HA | Sigma-Aldrich | Cat# H9658, RRID:AB_260092 |
| Mouse monoclonal anti-Cox2 | MitoScience | Cat# MS419, RRID:AB_1618187 |
| Rabbit polyclonal anti-Atg33 | Fukuda et al, 2023 | N/A |
| Rabbit polyclonal anti-Atg44 | Fukuda et al, 2023 | N/A |
| Goat anti-rabbit IgG-HRP | Jackson ImmunoResearch | Cat# 111-035-003 |
| Goat anti-mouse IgG-HRP | Jackson ImmunoResearch | Cat# 115-035-003 |
| **Chemicals, enzymes, and other reagents** | | |
| NEBuilder HiFi DNA Assembly Master Mix | New England Biolabs | Cat# E2621 |
| PrimeSTAR Mutagenesis Basal Kit | Takara Bio | Cat# R046A |
| EzWestLumi plus | ATTO | Cat# WSE-7120 |
| Clarity Max Western ECL Substrate | Bio-Rad | Cat# 1705062 |
| Zymolyase 100 T | Nacalai Tesque | Cat# 07665-55 |
| Proteinase K, recombinant, PCR Grade | Roche | Cat# 03115836001 |
| Amylose resin | New England Biolabs | Cat# E8021 |
| PreScission protease | Cytiva | Cat# 27-0843-01 |
| Alexa Fluor 488 $C_5$-maleimide | Invitrogen | Cat# A10254 |
| Alexa Fluor 647 $C_2$-maleimide | Invitrogen | Cat# A20347 |
| 1-palmitoyl-2-oleoyl-sn-glycero-3-phosphocholine (POPC) | Avanti Polar Lipids | Cat# 850457 |
| 1-palmitoyl-2-oleoyl-sn-glycero-3-phosphoethanolamine (POPE) | Avanti Polar Lipids | Cat# 850757 |
| L-α-phosphatidylinositol (PI) | Avanti Polar Lipids | Cat# 840042 |
| 1-palmitoyl-2-oleoyl-sn-glycero-3-phospho-L-serine (POPS) | Avanti Polar Lipids | Cat# 840034 |

| Reagent/resource | Reference or source | Identifier or catalog number |
|---|---|---|
| 1-palmitoyl-2-oleoyl-sn-glycero-3-phosphate (POPA) | Avanti Polar Lipids | Cat# 840857 |
| 1',3'-bis[1,2-dioleoyl-sn-glycero-3-phospho]-glycerol (CL) | Avanti Polar Lipids | Cat# 710335 |
| 1,2-dioleoyl-sn-glycero-3-phosphoethanolamine-N-(lissamine rhodamine B sulfonyl) (liss Rhod PE) | Avanti Polar Lipids | Cat# 810150 |
| PVA (MW 146,000–186,000) | Sigma-Aldrich | Cat# 363103 |
| **Software** | | |
| MetaMorph 7 | Molecular Devices | RRID:SCR_002368 |
| NIS-Elements | Nikon | RRID:SCR_014329 |
| Image Lab | Bio-Rad | RRID:SCR_014210 |
| Astra | Wyatt Technology | RRID:SCR_016255 |
| GROMACS (version 2022) | Abraham et al, 2015 | RRID:SCR_014565 |
| CHARMM-GUI | Monticelli et al, 2008 | RRID:SCR_025037 |
| PyMOL | Schrödinger, LLC | RRID:SCR_000305 |
| GraphPad Prim 10 (10.4.1) | GraphPad Software | RRID:SCR_002798 |

## Yeast strains

The *Saccharomyces cerevisiae* strains used in this study are listed in Table EV2. Gene deletion and tagging were performed as described previously (Longtine et al, 1998; Gueldener et al, 2002; Janke et al, 2004).

## Yeast culture conditions

Yeast cells were cultured at 30 °C in rich medium (YPD: 1% yeast extract, 2% peptone, and 2% glucose), lactate medium (YPL: 1% yeast extract, 2% peptone, and 2% lactate), or synthetic minimal medium with glucose (SMD: 0.67% yeast nitrogen base, 2% glucose, and amino acids) or lactate (SML: 0.67% yeast nitrogen base, 2% lactate, and amino acids). Nitrogen starvation experiments were performed in synthetic minimal medium lacking nitrogen (SD-N: 0.17% yeast nitrogen base without amino acids and ammonium sulfate, and 2% glucose).

## Fluorescence microscopy

Yeast cells expressing fluorescent proteins were cultured in the indicated media. Fluorescence images were captured using a Nikon Ti2 Eclipse microscope with a Plan Apo Lambda 100× oil objective lens and a CCD camera (MD-695, Molecular Devices) and analyzed using MetaMorph 7 (Molecular Devices) or NIS-Elements software (Nikon).

## Mitophagy assay

To monitor mitophagy, the Om45-GFP or Idh1-GFP processing assay was performed as previously described (Kanki and Klionsky,

2008; Kanki et al, 2009). In brief, cells were cultured in YPL or SML medium until mid-log phase, shifted to SD-N medium, and collected at the indicated time points. Cell lysates equivalent to $OD_{600} = 0.2$ units of cells were subjected to immunoblotting analysis.

## Plasmids

To construct Mfi2 and Mfi2(N66) expression plasmids under the control of the *CUP1* or *TDH3* promoter, their coding regions were amplified by PCR and cloned into the BamHI-XhoI sites of pCu416 (Labbé and Thiele, 1999) or pTDH416 (Fukuda et al, 2023). To construct AgMfi2 and DhMfi2 expression plasmids, their coding genes were artificially synthesized (Eurofins) and cloned into the BamHI-XhoI sites of pCu416. To construct a GST-fused truncated Mfi2 expression plasmid, the 26–99 a.a. coding region was amplified by PCR and cloned into the BamHI-XhoI sites of pGEX-4T-1 (Promega). Atg44 expression plasmids (pCu416-ATG44 and pTDH416-ATG44) were previously described (Fukuda et al, 2023). To construct Atg44 and Mfi2 expression plasmids for in vitro experiments, their coding genes were amplified by PCR and assembled into the downstream of the PreScission protease recognition sequence in a modified pET15b vector by using NEBuilder HiFi DNA assembly kit. A linker sequence (GlyGlyGlySerGlyGlyGlySer) was inserted between the maltose-binding protein (MBP) and PreScission protease recognition sequences. For fluorescent labeling, a cysteine residue was inserted into the linker sequence using PrimeSTAR Mutagenesis Basal Kit (TaKaRa).

## Antibodies

Anti-Mfi2 antibodies were produced by immunizing rabbits with recombinant GST-fused truncated Mfi2 (26–99 a.a.), followed by affinity purification with the same recombinant proteins transferred onto a polyvinylidene difluoride (PVDF) membrane (Merck Millipore). Anti-GFP (Takara Bio, 632380, RRID:AB_10013427), anti-Pgk1 (Thermo Fisher Scientific, 459250, RRID:AB_2532235), anti-Atp2 (Abcam, ab128743, RRID:AB_2810299), anti-HA (Sigma-Aldrich, H9658, RRID:AB_260092), anti-Cox2 (MitoScience, MS419, RRID:AB_1618187), anti-Atg33 (Fukuda et al, 2023), and anti-Atg44 (Fukuda et al, 2023) were used for immunoblotting.

## Immunoblotting analysis

Protein samples from yeast cells were resuspended in sodium dodecyl sulfate (SDS) sample buffer (50 mM Tris-HCl, pH 6.8, 10% glycerol, 2% SDS, 5% 2-mercaptoethanol, and 0.1% bromophenol blue), incubated at 42 °C for 1 h, and subjected to SDS-polyacrylamide gel electrophoresis (PAGE). Proteins were transferred from polyacrylamide gels to PVDF membranes using transfer buffer (25 mM Tris, pH 8.3, 192 mM glycine, 20% methanol). The membranes were blocked with phosphate-buffered saline (PBS) with Tween-20 (PBS-T; 10 mM $PO_4^{3-}$, pH 7.4, 140 mM NaCl, 2.7 mM KCl, and 0.05% Tween-20) containing 5% skim milk for 1 h. The membranes were incubated with primary antibodies in PBS-T containing 2% skim milk overnight at 4 °C and washed three times with PBS-T. The membranes were then incubated with secondary antibodies (Peroxidase-conjugated

AffiniPure Goat Anti-Rabbit IgG or Peroxidase-conjugated Affini-Pure Goat Anti-Mouse IgG) in PBS-T containing 2% skim milk for 1 h at room temperature and washed three times with PBS-T. Chemiluminescence signals were detected using EzWestLumi plus (ATTO) or Clarity Max Western ECL Substrate (Bio-Rad), captured with ChemiDoc XRS+ (Bio-Rad), and analyzed using Image Lab software (Bio-Rad).

## Isolation of mitochondria, sodium carbonate treatment, and ProK protection assay

Yeast cells were cultured in YPL until mid-log phase and resuspended in DTT buffer (10 mM dithiothreitol, 0.1 M Tris-HCl, pH 9.3) for 30 min at 30 °C. Cells were collected and converted to spheroplasts in sorbitol buffer (1.2 M sorbitol, 20 mM $KH_2PO_4$, pH 7.4) with Zymolyase 100 T (Nacalai Tesque). Spheroplasts were collected by centrifugation (1000×*g* for 10 min at 4 °C) and resuspended in ice-cold homogenization buffer (0.6 M sorbitol, 20 mM HEPES, pH 7.4) and homogenized in a Potter-Elvehjem homogenizer. The cell homogenate was centrifuged at 1000×*g* for 10 min at 4 °C. The supernatant was centrifuged at 6500×*g* for 10 min at 4 °C and the pellet was collected as the mitochondrial fraction. Isolated mitochondria were separately suspended in ice-cold 0.1 M sodium carbonate (pH 11.0), incubated for 30 min on ice, and then centrifuged at 100,000×*g* for 30 min at 4 °C. Proteins in the pellet and supernatant fractions were precipitated by adding 10% TCA. Isolated mitochondria were separately suspended in ice-cold homogenization buffer, hypotonic buffer (20 mM HEPES, pH 7.4), or hypotonic buffer with 0.5% Triton X-100 and treated with ProK (200 μg/ml) for 30 min on ice. The ProK reaction was stopped by adding 10% trichloroacetic acid (TCA). TCA-precipitated proteins were washed with acetone and subjected to immunoblotting.

## Preparation of recombinant Atg44 and Mfi2

Atg44, Mfi2, and Mfi2(C33) were overexpressed as MBP-fusion proteins in *Escherichia coli* C41(DE3) cells (New England Biolabs). Cells were grown in Luria-Bertani medium and protein expression was induced with 0.5 mM isopropyl-β-D-thiogalactopyranoside (IPTG). After culture at 30 °C for 3 h, cells were harvested by centrifugation and suspended in P buffer (PBS supplemented with 1.0 M NaCl). The suspension was supplemented with 1.0 mM phenylmethylsulfonyl fluoride and lysed by sonication. The supernatant of the lysate after centrifugation was loaded onto Amylose resin (New England Biolabs) pre-equilibrated with P buffer. The resin was washed extensively with P buffer and equilibrated with PBS buffer. The bound proteins were eluted with PBS buffer supplemented with 10 mM maltose, concentrated with a centrifugal device, Amicon Ultra-15 (Merck Millipore), and then subjected to SEC using Superose 6 Increase 10/300 GL column (Cytiva) in HEPES buffer (20 mM HEPES-NaOH, pH 7.0, and 150 mM NaCl). For the preparation of MBP and Mfi2(C33), the eluates from Amylose resin were treated with PreScission protease at 4 °C overnight, concentrated with a centrifugal device, and subsequently subjected to SEC using Superdex 30 Increase 10/300 GL column (Cytiva) in HEPES buffer. The eluates were concentrated with a centrifugal device and frozen at −80 °C for storage.

## SEC-MALS

SEC-MALS was conducted with a chromatography system connected in-line to a DAWN HELEOS II (Wyatt) for light scattering and RI-501 (Shodex) for differential refractive index measurements. 100 µl of 50 µM MBP-Atg44 or MBP-Mfi2 was subjected to SEC using Superose 6 Increase 10/300 GL column in HEPES buffer at room temperature. Molar masses of elution peaks were calculated with Astra software (Wyatt) using the values of light scattering, differential refractive index, and ultraviolet absorption at 280 nm.

## Fluorescent labeling

Cys-introduced MBP-Atg44 was mixed with Alexa Fluor 488 $C_5$-maleimide (Invitrogen) dissolved in dimethyl sulfoxide (DMSO) at an equivalent molar ratio. Cys-introduced MBP-Mfi2 was mixed with Alexa Fluor 647 $C_2$-maleimide (Invitrogen) dissolved in DMSO at an equivalent molar ratio. The mixture was incubated for 30 min at room temperature and dialyzed against HEPES buffer using a mini dialysis kit (Cytiva) at 4 °C overnight to remove remaining fluorescent dyes.

## Membrane binding experiments

1-palmitoyl-2-oleoyl-sn-glycero-3-phosphocholine (PC), 1-palmitoyl-2-oleoyl-sn-glycero-3-phosphoethanolamine (PE), L-α-phosphatidylinositol (PI), 1-palmitoyl-2-oleoyl-sn-glycero-3-phospho-L-serine (PS), 1-palmitoyl-2-oleoyl-sn-glycero-3-phosphate (PA), 1',3'-bis[1,2-dioleoyl-sn-glycero-3-phospho]-glycerol (CL) and 1,2-dioleoyl-sn-glycero-3-phosphoethanolamine-N-(lissamine rhodamine B sulfonyl) (liss Rhod PE) were purchased from Avanti Polar Lipids. GUVs were prepared by a gel-assisted swelling method as described elsewhere (Zhu et al, 2024). Briefly, a thin layer of 5% poly(vinyl alcohol) (PVA) with an average molecular weight of 146,000–186,000 (Sigma-Aldrich) dissolved in water was applied on a coverslip pre-cleaned with water and ethanol and dried at 50 °C. 1.0 mM lipid solution in chloroform was applied to the PVA sheet and dried in a desiccator connected to a vacuum pump for 1 h. The dried lipid film on the PVA sheet was then hydrated in 320 mM sucrose for at least 30 min at room temperature to produce GUVs. The solution and PVA sheet were transferred to a plastic tube, and the PVA sheet was removed before the solution containing GUVs was used.

For membrane binding analysis, a small aliquot of the solution containing GUVs was added to HEPES buffer in a glass-bottom dish (MatTek) passivated with BSA (Wako). Fluorescently labeled MBP-Atg44 or MBP-Mfi2 were then added to the solution at final concentrations of 2.0 µM and mixed gently. Confocal images were acquired at room temperature with a confocal laser scanning microscope, A1 LFOV (Nikon) with a 60× oil immersion objective lens (Nikon) using NIS-Elements software (Nikon).

## Tube fission experiments

Lipid nanotubes were prepared using excess membrane reservoirs as described previously (Fukuda et al, 2023; Maruyama and Noda, 2023). Fluorescently-labeled MBP-Atg44, MBP-Mfi2, non-labeled

Mfi2(C33), or MBP was applied to lipid nanotubes at final concentrations of 2.0 µM in HEPES buffer in an eight-well chamber cover (Matsunami Glass) passivated moderately with BSA. Confocal images were acquired at room temperature with A1 LFOV confocal microscope with a 60× oil immersion objective lens using NIS-Elements software. Tube diameter was estimated from fluorescence intensity as described previously (Fukuda et al, 2023; Maruyama and Noda, 2023).

## Coarse-grained molecular dynamics simulations

Simulations of Mfi2 on membranes were performed using a coarse-grained model with the GROMACS (version 2022) (Abraham et al, 2015) and the MARTINI model (version 2.2) (Marrink et al, 2007; Monticelli et al, 2008). The initial structure of the protein was predicted using AlphaFold3 (Abramson et al, 2024). The initial position of the protein was set 3 nm above the phosphorus atoms on the membrane surface, oriented parallel to the membrane. The tertiary structure of the protein in the coarse-grained model was maintained using an elastic network model. Each leaflet of the membrane was set to a composition of 200 lipid molecules with a ratio of POPC:POPE:POCL2 = 6:2:2 or POPC:POPE = 8:2. The proteins and lipids were solvated in a 20 nm × 20 nm × 20 nm box with coarse-grained waters and 0.15 M NaCl ions. The initial configurations were built by the Martini Maker module in CHARMM-GUI server (Monticelli et al, 2008). Each system was first energy minimized and equilibrated using the Berendsen thermostat and barostat (Berendsen et al, 1984) followed by 1 µs production runs with a 20-fs time step. System temperature and pressure during the production phase were maintained at 303.15 K and 1 atm with the velocity rescaling thermostat and the semi-isotropic Parrinello–Rahman barostat (Parrinello and Rahman, 1981), respectively. The simulation results were visualized using PyMOL (http://www.pymol.org/pymol).

## Statistical analyses

Statistical analyses were performed using GraphPad Prism Version 10.4.1 (GraphPad Software). Statistical significance was determined using unpaired two-tailed Welch's $t$ test. Data are presented as mean ± standard deviation (SD) from at least three independent experiments. $P$ values < 0.05 were considered statistically significant.

# Data availability

This study includes no data deposited in external repositories.

The source data of this paper are collected in the following database record: biostudies:S-SCDT-10_1038-S44319-025-00689-x.

# Peer review information

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

## Acknowledgements

This study was supported in part by the Japan Society for the Promotion of Science grants JP23H04255 and JP24K01717 (to KF), JP24K09373 (to TM), JP23K05715, JP24K01980, and JP25K09559 (to YS), JP23K20044, JP24H00060, JP25H00966, JP25H01320, and JP25H01321 (to NNN), JP23K23878, JP24H02274, JP25K21747, and JP25H01101 (to TK); AMED-CREST grant JP23gm1710006s0101 (to TK); JST-CREST grant JPMJCR20E3 (to NNN); the Takeda Science Foundation (to KF); Institute for Fermentation, Osaka (IFO) (to KF); the Noda Institute of Scientific Research (to KF). The numerical computations were performed with the supercomputer systems Fugaku (hp230252 and hp250019) and HOKUSAI at RIKEN, Wisteria/BDEC-01 at the University of Tokyo. We thank Edanz (https://jp.edanz.com/) for editing a draft of this manuscript.

## Author contributions

Kentaro Furukawa: Conceptualization; Data curation; Formal analysis; Supervision; Funding acquisition; Validation; Investigation; Visualization; Methodology; Writing—original draft; Writing—review and editing. Tatsuro Maruyama: Conceptualization; Data curation; Formal analysis; Funding acquisition; Validation; Investigation; Visualization; Methodology; Writing—original draft; Writing—review and editing. Yuji Sakai: Data curation; Formal analysis; Funding acquisition; Validation; Investigation; Visualization; Methodology; Writing—original draft; Writing—review and editing. Shun-ichi Yamashita: Investigation; Writing—review and editing. Keiichi Inoue: Investigation; Writing—review and editing. Tomoyuki Fukuda: Investigation; Writing—review and editing. Nobuo N Noda: Conceptualization; Resources; Supervision; Funding acquisition; Methodology; Writing—original draft; Project administration; Writing—review and editing. Tomotake Kanki: Conceptualization; Resources; Supervision; Funding acquisition; Validation; Methodology; Writing—original draft; Project administration; Writing—review and editing.

Source data underlying figure panels in this paper may have individual authorship assigned. Where available, figure panel/source data authorship is listed in the following database record: biostudies:S-SCDT-10_1038-S44319-025-00689-x.

## Disclosure and competing interests statement

The authors declare no competing interests.

# Expanded View Figures

**Figure EV1.   Identification of Mco12/Mfi2 as an Atg44-like protein.**

(A) The indicated cells were cultured in YPL until mid-log phase and shifted to SD-N. Cells were collected after 6 h, and Om45-GFP processing was monitored by immunoblotting. Pgk1 was detected as a loading control. (B) The indicated cells were cultured in YPL until mid-log phase, and their mitochondrial morphology was analyzed by fluorescence microscopy. More than 300 cells were analyzed, and representative images are shown. Scale bar, 4 µm. (C) Protein sequence alignment of Atg44 and Mco12/Mfi2. Asterisk, conserved residue; colon, residue with highly similar properties; period, residue with weakly similar properties. (D) AlphaFold-predicted structures of Atg44 and Mco12/Mfi2. N and C indicate the N- and C-termini, respectively. (E) TMHMM (https://services.healthtech.dtu.dk/services/TMHMM-2.0/)-predicted hydrophobic regions of Atg44 and Mco12/Mfi2, showing weakly hydrophobic regions near the N-terminus of Mco12/Mfi2. Source data are available online for this figure.

▶

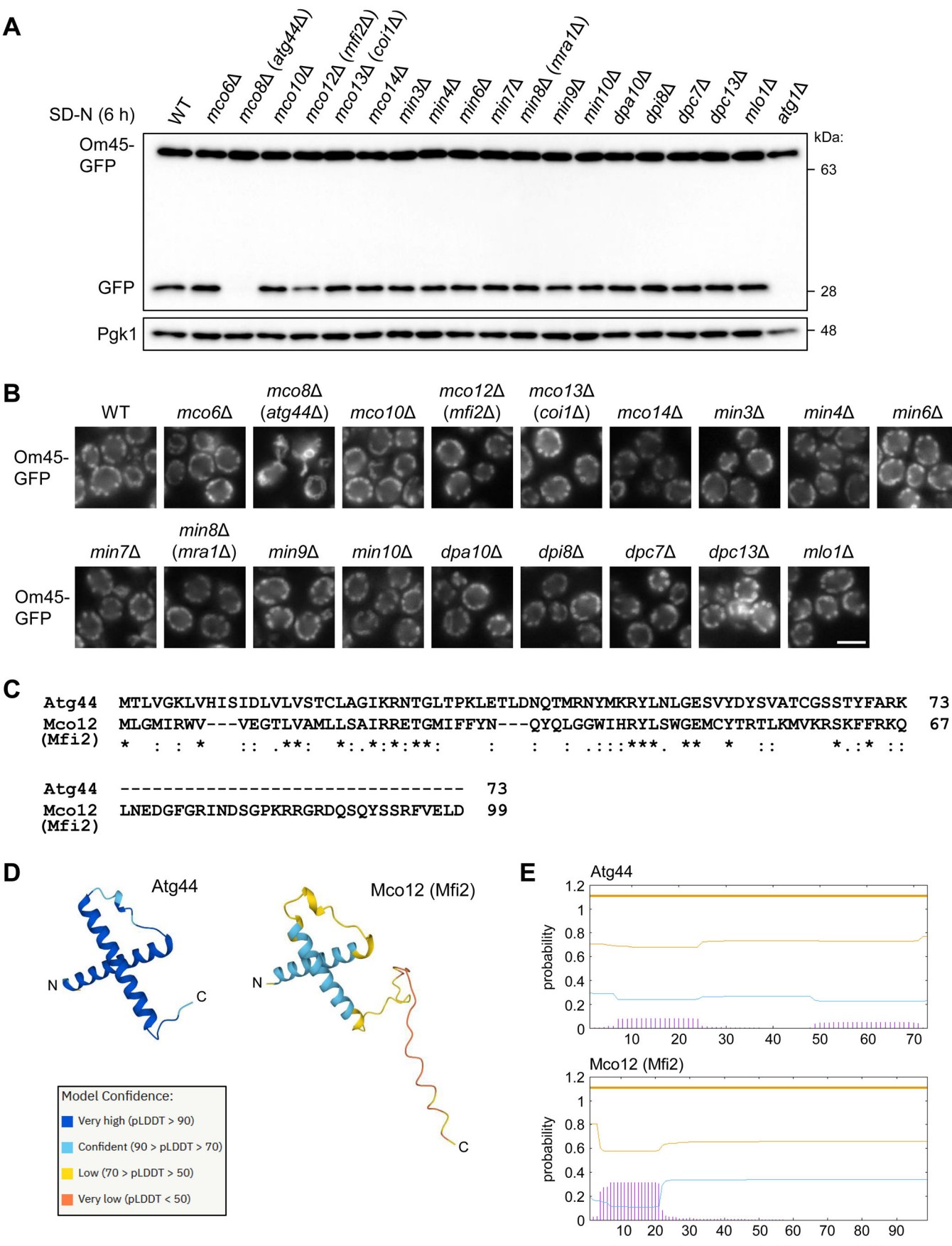

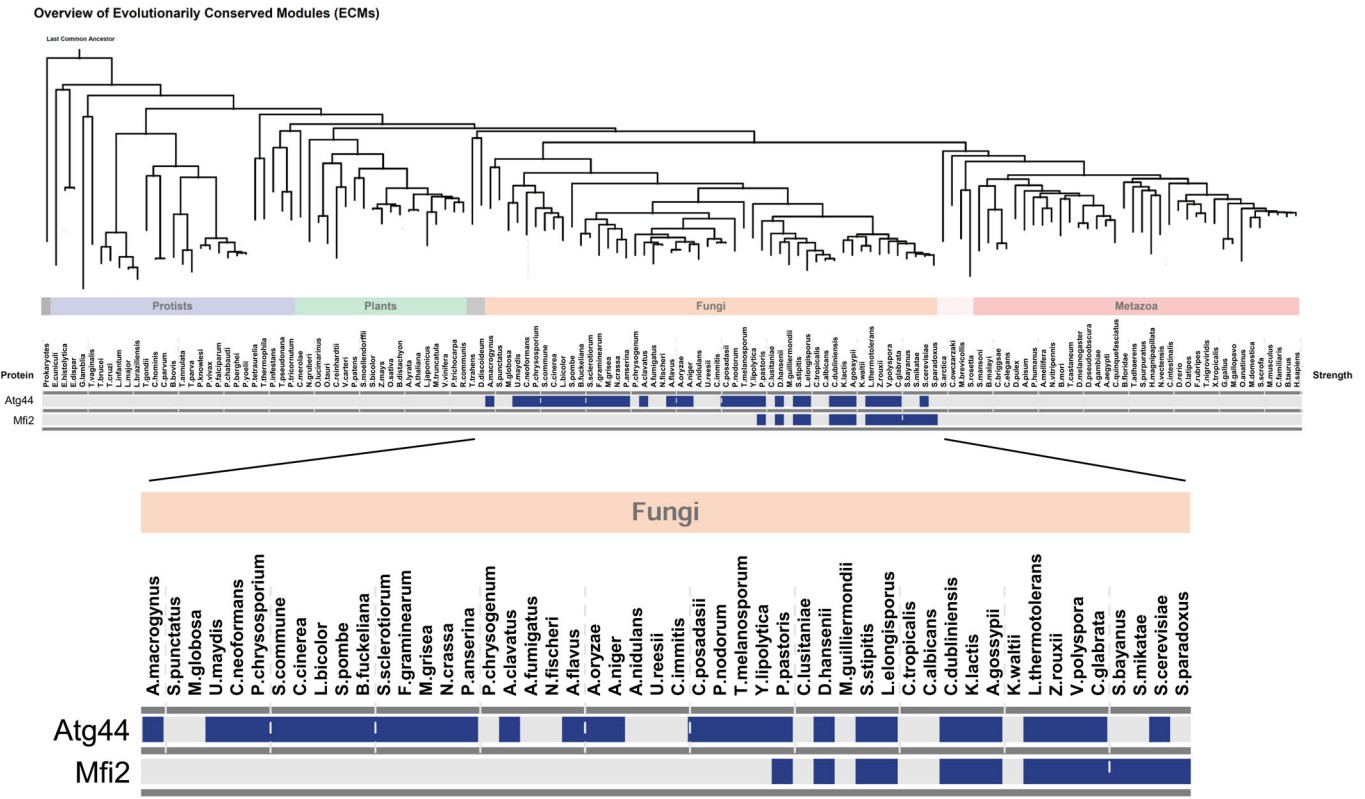

**Figure EV2. CLIME analysis of Atg44 and Mfi2.**

CLIME analysis (https://www.gene-clime.org/) shows less conservation of Mfi2 than of Atg44 among fungi.

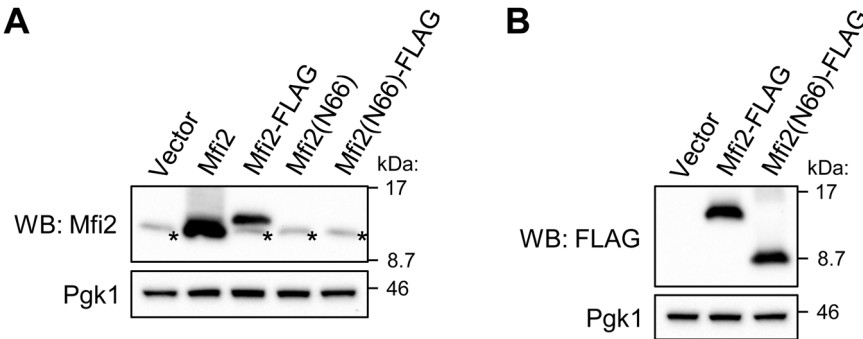

**Figure EV3. Expression levels of Mfi2 constructs.**

(A) *atg44Δ* cells expressing the indicated proteins under the *TDH3* promoter were cultured in SML medium until mid-log phase, and cell lysates were analyzed by immunoblotting. Endogenous Mfi2 and overexpressed Mfi2 were detected with anti-Mfi2 antibody, whereas overexpressed Mfi2(N66) was not detected because it lacks the C-terminal region recognized by the antibody. Asterisks indicate endogenous Mfi2. (B) Overexpressed Mfi2-FLAG and Mfi2(N66)-FLAG were detected at comparable levels with anti-FLAG antibody. These analyses demonstrate that (i) overexpressed Mfi2 was present at much higher levels than endogenous Mfi2, and (ii) Mfi2-FLAG and Mfi2(N66)-FLAG were expressed at similar levels. Pgk1 was used as a loading control. Source data are available online for this figure.

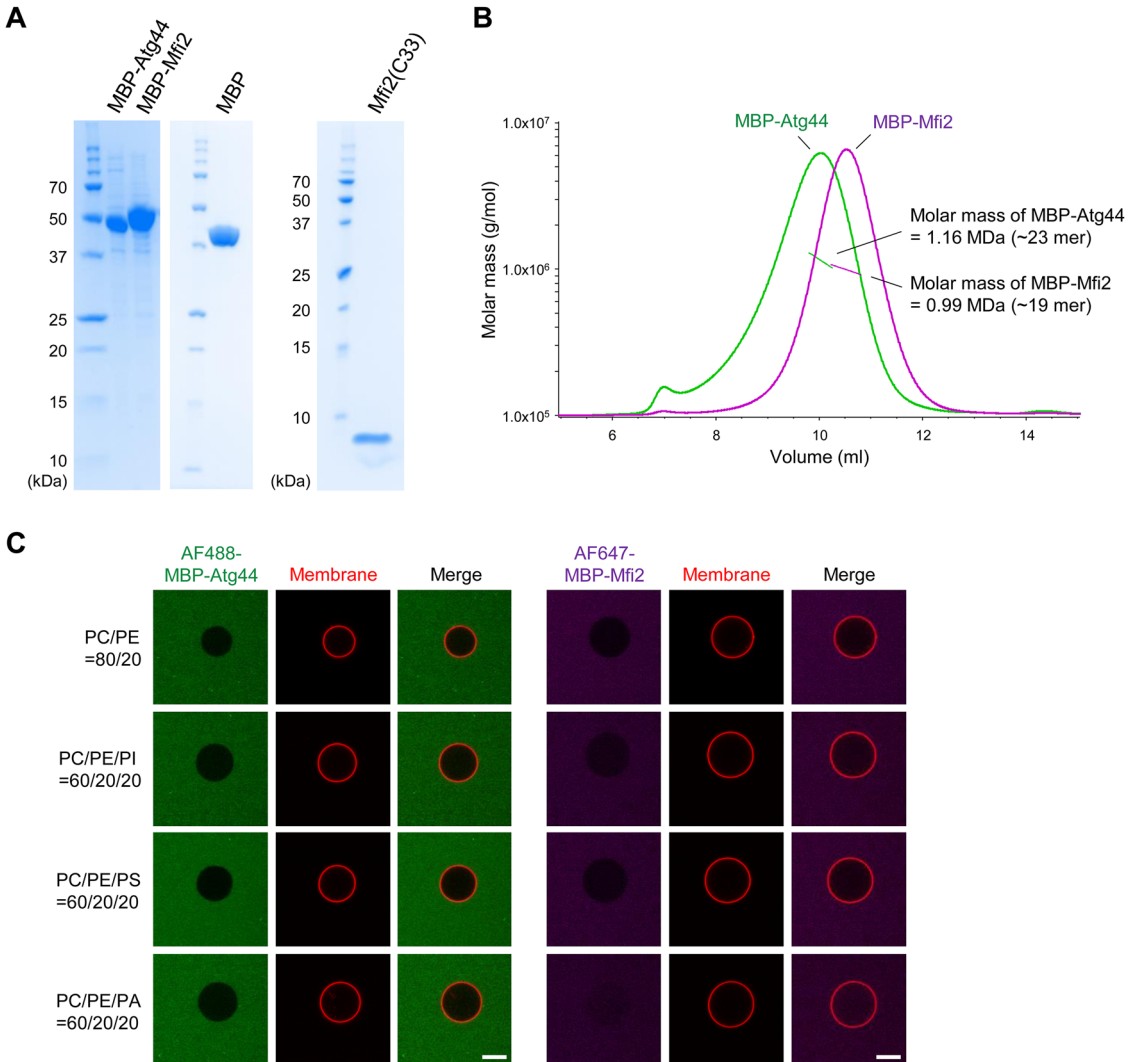

**Figure EV4. Sample preparation and membrane binding.**

(A) SDS-PAGE analysis of the purity of prepared samples. The gel was stained with Coomassie Brilliant Blue. (B) SEC-MALS analysis of the oligomerization states of MBP-Atg44 and MBP-Mfi2. Absorbance at 280 nm in SEC was shown as solid line. Estimated molar mass was shown as dot. (C) Membrane binding of MBP-Atg44 and MBP-Mfi2. Membrane binding was examined by confocal laser scanning microscopy using fluorescently labeled proteins and GUVs labeled with liss Rhod PE. Scale bars, 10 μm. Source data are available online for this figure.

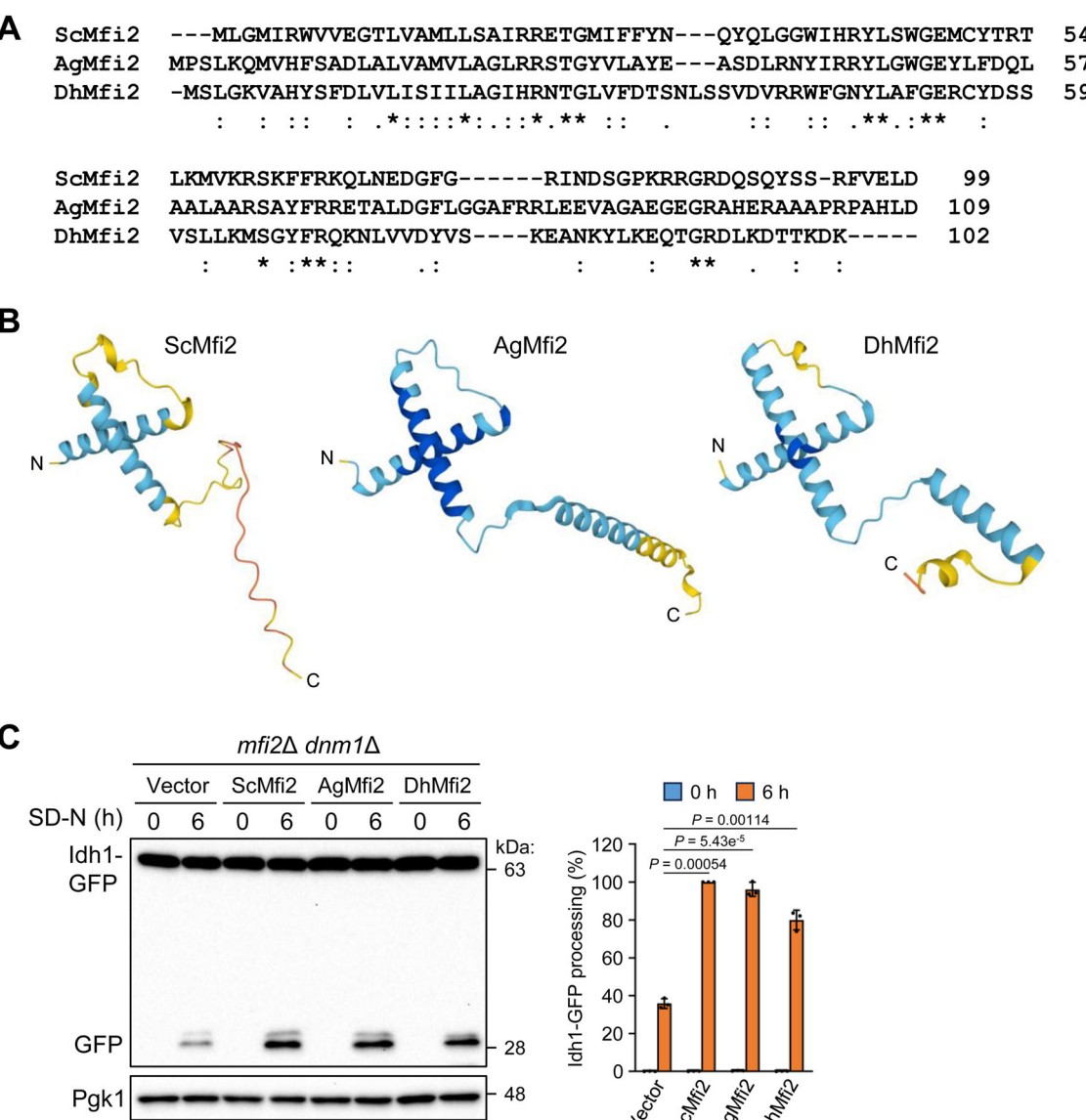

**Figure EV5. Exogenous expression of AgMfi2 or DhMfi2 rescues the impaired mitophagy in *dnm1Δ mfi2Δ* cells.**

(A) Protein sequence alignment of Mfi2 homologs. Sc, *Saccharomyces cerevisiae*; Ag, *Ashbya gossypii*; Dh, *Debaryomyces hansenii*. Asterisk, conserved residue; colon, residues with highly similar properties; period, residues with weakly similar properties. (B) AlphaFold2-predicted structures of Mfi2 homologs. (C) The *dnm1Δ mfi2Δ* cells expressing the indicated proteins were cultured in SML until mid-log phase and shifted to SD-N. Cells were collected at the indicated time points, and Idh1-GFP processing was monitored by immunoblotting. The value of ScMfi2 (6 h) was set to 100%. The quantification results are shown as mean ± SD (*n* = 3 biological replicates). Welch's *t* test; exact *P* values are shown in the graph. Source data are available online for this figure.

