## [Peer Review File · EMBO Reports]

Mitochondrial fission during mitophagy requires both inner and outer mitofissins

Kentaro Furukawa, Tatsuro Maruyama, Yuji Sakai, Shun-ichi Yamashita, Keiichi Inoue, Tomoyuki Fukuda, Nobuo Noda, and Tomotake Kanki

Corresponding author(s): Tomotake Kanki (kanki.tomotake.114@m.kyushu-u.ac.jp), Kentaro Furukawa (furukawa.kentaro.828@m.kyushu-u.ac.jp)

Review Timeline:

Transfer Date:	16th Jun 25
Editorial Decision:	26th Jun 25
Revision Received:	17th Oct 25
Editorial Decision:	4th Dec 25
Revision Received:	8th Dec 25
Accepted:	15th Dec 25

Transaction Report: This manuscript was transferred to EMBO reports following peer review at Review Commons.

**Review
COMMONS**

Review #1

1. Evidence, reproducibility and clarity:

Evidence, reproducibility and clarity (Required)

Furukawa and colleagues identified Mfi2 as novel factor that promotes fragmentation and removal of damaged mitochondria by mitophagy. They report that parallel loss of Dnm1 and Mfi2 blocks mitophagy. Mfi2 acts on the outer membrane, while the previous found Atg44 functions in the intermembrane space. How the proteins cooperate remains unknown. This is an elegant study with high-quality data. The findings are interesting for a broad readership. There are some issues as outline below that should be solved.

1. It remains unclear how Mfi2 is anchored into the outer mitochondrial membrane. Does it contain a transmembrane domain? The carbonate resistance indicates the presence of such transmembrane domain. However, the presented structures lack such membrane-spanning segment. This point should be clarified.
2. How does Mfi2 cooperate with Dnm1? Is there any interaction between these proteins? Some further information could provide mechanistic insights into the function of Mfi2.
3. The authors report a CL-dependent binding of Mfi2 to liposomes. Is the recruitment of Mfi2 to mitochondria impaired when CL-synthesis is blocked, e.g. in *crd1delta* mitochondria?
4. Figure 4B: a wild-type control should be added.

2. Significance:

Significance (Required)

The reported findings are interesting for a broad readership.

3. How much time do you estimate the authors will need to complete the suggested revisions:

Estimated time to Complete Revisions (Required)

(Decision Recommendation)

Between 1 and 3 months

4. Review Commons values the work of reviewers and encourages them to get credit for their work. Select 'Yes' below to register your reviewing activity at Web of Science Reviewer Recognition Service (formerly Publons); note that the content of your review will not be visible on Web of Science.

No

Review #2

1. Evidence, reproducibility and clarity:

Evidence, reproducibility and clarity (Required)

In this study, the authors discover a mitochondrial fission factor, termed Mfi2, that promotes mitophagy efficiency and that functions in a partially redundant way with Dnm1 for the fission of mitochondrial outer membranes during mitophagy. The discovery helps to clarify why Dnm1 does not appear to be essential for fission mediated mitophagy by Dnm1. Mfi2 is structurally similar to the inner membrane fission factor Atg44 which is consistent with Mfi2's fission activity. The authors show that Mfi2 has membrane fission activity towards nanotubes in vitro, and that membrane binding is dependent of high levels of cardiolipin, a mitochondrially enriched lipid. In summary, the authors show that Mfi2 mediates mitochondrial outer membrane fission together with Drp1, whereas Atg44 mediates inner membrane fission, that together are necessary for mitophagy.

****Major:****

1. Figure 2: How do the expression levels of the Mfi2 constructs compare to the endogenous levels of the protein? This will help to gauge to what degree Mfi2 N66 overexpression is needed to achieve mitochondrial fragmentation in Atg44 delta cells and also the low level of mitophagy rescue that was observed.
2. Figure 3A-B: The cardiolipin binding results in vitro are interesting but the concentration of cardiolipin is much lower on the outer membrane versus the inner membrane. Can the authors comment on whether the cardiolipin levels used on the nanotubes are relevant to that of the mitochondrial outer membrane where Mfi2 is located? Can the authors provide quantitative data for these experiments to help strengthen their conclusions? Can the authors also use purified MBP alone or a form of Mfi2 that cannot bind to membrane e.g. Mfi2-C33) as a control?
3. Figure 4D: The protrusions are very difficult to visualize. Can the authors also provide zoomed in regions. Is the data representative from 3 or more independent experiments? Can the authors provide a graph of the quantitation to aid readers with analysis of the data?

4. Figure 4D: It is fascinating to see the mitochondrial protrusion formation being dependent on autophagy factors but not mitochondrial fission factors. To help visualize this, can the authors image one of either Atg1, Atg8 to address whether phagophores are forming on the protrusions and if so where they are positionally located on the protrusion in control and/or mfi2,dnm1,atg44 triple mutant cells?

****Minor:****

1. Is it possible to target Atg44 to the mitochondrial outer membrane, either by attaching an OM anchor or using part of the N-terminus of Mfi2? This will help elucidate how Mfi2 reaches the outer membrane and whether Atg44 can be just as active on the outer membrane as long as it can access it.

2. Are microtubules or actin required for the protrusions to form? Using the triple mutant cells that have a high proportion of protrusions, it could be tried to add cytoskeletal depolymerizing drugs such as nocodazole for microtubules or Latrunculin A or Latrunculin B for actin.

2. Significance:

Significance (Required)

The discovery of Mfi2 as an outer membrane mitophagy fission factor is an exciting, and very important and significant contribution to the field. The data in this study are clear and the conclusions are generally well supported by the experiments. This study appears to be suitable as a report style manuscript given that there is limited mechanistic analysis of Mfi2 activity. This does not affect the importance of the work, it just means that it is suited as a report of a significant discovery. Overall, this fills an important knowledge gap in solving the mystery behind which factors are involved in mitochondrial outer membrane fission during mitophagy, and provides a clarification why Dnm1 loss alone minimally affects mitophagy. This work will appeal to researchers interested in mitochondrial biology, the autophagy field, and cell biologists interested in organelle membrane dynamics, and is also broadly important and interesting to all cell biologists.

Reviewer expertise: mitophagy mechanisms, cell biology of mitophagy, autophagy and autophagosome formation, mitochondrial biology including OXPHOS and mitochondrial dynamics

3. How much time do you estimate the authors will need to complete the suggested revisions:

Estimated time to Complete Revisions (Required)

(Decision Recommendation)

Between 3 and 6 months

Yes

Review #3

1. Evidence, reproducibility and clarity:

Evidence, reproducibility and clarity (Required)

The manuscript by Furukawa et al. presents a well-structured and thorough study identifying Mfi2 as a novel mitochondrial outer membrane-resident fission factor required for mitophagy in *Saccharomyces cerevisiae*. The authors demonstrate that Mfi2, together with the inner membrane mitofission Atg44 and the dynamin-related GTPase Dnm1, contributes to mitochondrial fragmentation during mitophagy. Importantly, they show that while Dnm1 is dispensable on its own, Mfi2 and Dnm1 act redundantly from the outer membrane to support Atg44-mediated fission. The data are robust, the figures are clear, and the mechanistic insight into how mitophagy-specific fission is achieved is of high relevance to the field of mitochondrial quality control.

Overall, this is a logically constructed and convincing study with important implications for understanding compartment-specific mechanisms of mitochondrial fission during selective autophagy. The conclusions are largely well supported by the data. However, a few issues and points of clarification should be addressed before publication.

****Major Comments****

1. The observation that both Mfi2 and Atg44 require high cardiolipin (CL) content for membrane binding and fission in vitro is intriguing, especially given that CL is enriched in

the inner membrane. The authors mention CL externalisation during mitophagy, but this connection could be made more explicit earlier in the manuscript. Furthermore, since the molecular mechanism of membrane interaction remains unresolved, I would strongly encourage the authors to undertake coarse-grained molecular dynamics simulations to explore how Mfi2 might interact with lipid bilayers of differing composition. This could clarify the role of CL and the potential structural contribution of the disordered C-terminal region.

2. While the genetic and phenotypic data indicate that Mfi2 and Dnm1 act independently to support mitochondrial fission, the spatial and temporal organisation of their activity during mitophagy remains unclear. Do Mfi2 and Dnm1 colocalise at fission sites, or do they act at separate subdomains of the outer membrane? Live-cell imaging with fluorescently tagged Mfi2 and Dnm1, particularly during mitophagy induction, could help clarify whether these factors act in concert or at distinct locations and time points. This would also help determine whether their apparent redundancy reflects parallel mechanisms or functional compensation at shared sites. It would also be interesting to combine this with Atg44.

****Minor Comments****

1. The sodium carbonate extraction and proteinase K assays (Figure 1E-F) are standard but may not be familiar to all readers. A brief explanatory sentence clarifying what these methods reveal about membrane topology would improve accessibility.

2. While immunoblot quantifications are shown throughout, it would be helpful to include statistical analysis where appropriate, especially in cases where differences between genotypes or constructs are modest.

3. The naming of Mfi2 as a mitofissin is consistent with previous terminology introduced for Atg44, but the term remains relatively new. A brief clarification distinguishing "mitofissin" from the better-known "mitofusin" family in mammals would help avoid confusion for readers less familiar with yeast-specific nomenclature.

2. Significance:

Significance (Required)

This is a strong and well-executed study that provides mechanistic insight into how mitochondrial fission is coordinated during mitophagy in yeast. A major strength is the identification and characterisation of Mfi2 as a previously unrecognised outer membrane fission factor acting in parallel with Dnm1 and in coordination with the intermembrane space protein Atg44. The genetic, imaging, and in vitro biochemical data are carefully integrated, and the authors are transparent about limitations, including open questions

around the C-terminal domain of Mfi2, CL dependence, and the evolutionary conservation of mitofissins.

The work makes a conceptual advance by showing that mitophagy-specific mitochondrial fission requires the cooperation of spatially separated factors acting from both the inside and outside of mitochondria, a mechanism that had not been fully appreciated. This study helps resolve previous contradictions regarding the dispensability of Dnm1 in mitophagy, thereby filling a gap in our understanding of organelle-specific fission. While the findings are focused on yeast, they raise broader questions about whether similar principles apply to higher eukaryotes (historically yeast research was always at the forefront of autophagy field).

The study will be of interest to specialists in autophagy, mitochondrial dynamics, and yeast cell biology, as well as researchers working on membrane remodelling and organelle quality control. While the audience is primarily specialised, the conceptual insights will resonate more broadly in the cell biology community.

I am an expert in mitophagy mechanisms in mammalian cells, and while not a specialist in yeast models, I found the study logical, rigorous, and of clear relevance to the broader autophagy field.

3. How much time do you estimate the authors will need to complete the suggested revisions:

Estimated time to Complete Revisions (Required)

(Decision Recommendation)

Between 3 and 6 months

Yes

Dear Prof. Kanki

Thank you for the submission of your manuscript to EMBO reports. I have now read your manuscript, the referee reports and your point-by-point response. I agree that your study might make an interesting contribution to our journal and would therefore like to invite you to revise your study along the lines suggested in your revision plan. Your revised manuscript will then be sent back to the same referees.

Acceptance of the manuscript will depend on a positive outcome of a second round of review. It is EMBO Reports policy to allow a single round of revision only and acceptance or rejection of the manuscript will therefore depend on the completeness of your responses included in the next, final version of the manuscript.

We realize that it is difficult to revise to a specific deadline. In the interest of protecting the conceptual advance provided by the work, we recommend a revision within 3 months (September 26th). Please discuss the revision progress ahead of this time with the editor if you require more time to complete the revisions.

I am also happy to discuss the revision further via e-mail or a video call, if you wish.

Your study would be published as a short report. For short reports, the revised manuscript should not exceed 27,000 characters (including spaces but excluding materials & methods and references) and 5 main plus 5 expanded view figures. The results and discussion sections must further be combined, which will help to shorten the manuscript text by eliminating some redundancy that is inevitable when discussing the same experiments twice. The entire materials and methods must be included in the main manuscript file.

=====
IMPORTANT NOTE:

We perform an initial quality control of all revised manuscripts before re-review. Your manuscript will FAIL this control and the handling will be delayed IN CASE the following APPLIES:

- 1) A data availability section providing access to data deposited in public databases is missing. If you have not deposited any data, please add a sentence to the data availability section that explains that.
- 2) Your manuscript contains statistics and error bars based on $n=2$. Please use scatter blots in these cases. No statistics should be calculated if $n=2$.

=====
When submitting your revised manuscript, we will require:

- 1) a .docx formatted version of the manuscript text (including legends for main figures, EV figures and tables). Please make sure that the changes are highlighted to be clearly visible.
- 2) individual production quality figure files as .eps, .tif, .jpg (one file per figure). Please download our Figure Preparation Guidelines (figure preparation pdf) from our Author Guidelines pages <https://www.embopress.org/page/journal/14693178/authorguide> for more info on how to prepare your figures.
- 3) a .docx formatted letter INCLUDING the reviewers' reports and your detailed point-by-point responses to their comments. As part of the EMBO Press transparent editorial process, the point-by-point response is part of the Review Process File (RPF), which will be published alongside your paper.
- 4) a complete author checklist, which you can download from our author guidelines (). Please insert information in the checklist that is also reflected in the manuscript. The completed author checklist will also be part of the RPF.
- 5) Please note that all corresponding authors are required to supply an ORCID ID for their name upon submission of a revised manuscript (). Please find instructions on how to link your ORCID ID to your account in our manuscript tracking system in our Author guidelines
()

6) We replaced Supplementary Information with Expanded View (EV) Figures and Tables that are collapsible/expandable online. A maximum of 5 EV Figures can be typeset. EV Figures should be cited as 'Figure EV1, Figure EV2' etc... in the text and their respective legends should be included in the main text after the legends of regular figures.

7) Please note that a Data Availability section at the end of Materials and Methods is now mandatory. In case you have no data that requires deposition in a public database, please state so instead of refereeing to the database. See also < <https://www.embopress.org/page/journal/14693178/authorguide#dataavailability>>. Please note that the Data Availability Section is restricted to new primary data that are part of this study.

Additional information on source data and instruction on how to label the files are available

10) Figure legends and data quantification:

- the name of the statistical test used to generate error bars and P values,
 - the EXACT p-values,
 - the number (n) of independent experiments (please specify technical or biological replicates) underlying each data point,
 - the nature of the bars and error bars (s.d., s.e.m.)
- If the data are obtained from n {less than or equal to} 5, show the individual data points in addition to the SD or SEM.
- If the data are obtained from n {less than or equal to} 2, use scatter blots showing the individual data points.

11) Our journal encourages inclusion of *data citations in the reference list* to directly cite datasets that were re-used and obtained from public databases. Data citations in the article text are distinct from normal bibliographical citations and should directly link to the database records from which the data can be accessed. In the main text, data citations are formatted as follows: "Data ref: Smith et al, 2001" or "Data ref: NCBI Sequence Read Archive PRJNA342805, 2017". In the Reference list, data citations must be labeled with "[DATASET]". A data reference must provide the database name, accession number/identifiers and a resolvable link to the landing page from which the data can be accessed at the end of the reference. Further instructions are available at .

12) All Materials and Methods need to be described in the main text using our 'Structured Methods' format. According to this format, the Methods section includes a Reagents and Tools Table (listing key reagents, experimental models, software and relevant equipment and including their sources and relevant identifiers) followed by a Methods and Protocols section describing the methods, ideally using a step-by-step protocol format. The aim is to facilitate adoption of the methodologies across labs. Please download and fill our Reagents and Tools Table template (.docx), which you can find in our author guidelines:

13) As part of the EMBO publication's Transparent Editorial Process, EMBO Reports publishes online a Review Process File to accompany accepted manuscripts. This File will be published in conjunction with your paper and will include the referee reports, your point-by-point response and all pertinent correspondence relating to the manuscript.

Yours sincerely,

Full Revision

Manuscript number: RC-2025-03004 (EMBOR-2025-62137V1-T)

Corresponding author(s): Kentaro Furukawa and Tomotake Kanki

1. General Statements [optional]

We would like to thank the reviewers for their constructive and positive feedback. We are encouraged that all three reviewers consider the identification of Mfi2 as an outer mitochondrial membrane fission factor required for mitophagy to be a significant and important contribution to the research field. We have carefully addressed all the concerns raised through additional experiments, clarifications, and textual revisions. We believe that these revisions have further strengthened the manuscript and enhanced its impact.

Reviewer #1 (Evidence, reproducibility and clarity (Required)):

Furukawa and colleagues identified Mfi2 as novel factor that promotes fragmentation and removal of damaged mitochondria by mitophagy. They report that parallel loss of Dnm1 and Mfi2 blocks mitophagy. Mfi2 acts on the outer membrane, while the previous found Atg44 functions in the intermembrane space. How the proteins cooperate remains unknown. This is an elegant study with high-quality data. The findings are interesting for a broad readership. There are some issues as outline below that should be solved.

Response:

We would like to thank Reviewer #1 for their thoughtful evaluation of our manuscript and for recognizing the interest and quality of the study.

1. It remains unclear how Mfi2 is anchored into the outer mitochondrial membrane. Does it contain a transmembrane domain? The carbonate resistance indicates the presence of such transmembrane domain. However, the presented structures lack such membrane-spanning segment. This point should be clarified.

Response:

We performed an *in silico* topology prediction of Atg44 and Mfi2 using TMHMM. This tool identified a weakly hydrophobic region of Mfi2 near the N-terminus but did not predict a definitive transmembrane domain (new Fig. EV1E). This result implies that Mfi2 interacts with the outer membrane in a monotopic or peripheral manner, rather than as a classical transmembrane protein. Such proteins may remain in the membrane pellet after carbonate treatment due to their strong hydrophobic insertion into the lipid bilayer (e.g., yeast tafazzin/Taz1; Brandner *et al.*, *Mol Biol Cell*, 2005). We have incorporated this interpretation into the revised manuscript (Page 6, lines 10-13).

2. How does Mfi2 cooperate with Dnm1? Is there any interaction between these proteins? Some further information could provide mechanistic insights into the function of Mfi2.

Response:

While our study does not explicitly claim that Mfi2 cooperates with Dnm1, we investigated whether these proteins physically associate. We performed co-immunoprecipitation experiments with Mfi2-FLAG and Dnm1-GFP under both growing and mitophagy-inducing conditions, but we did not detect any interaction between these proteins. As a positive control, co-immunoprecipitation of endogenous Mfi2 with Mfi2-FLAG was readily detected in the same experimental setup, similar to what we previously observed for Atg44-FLAG interacting with endogenous Atg44 (our unpublished data), confirming that our assay was functional (Response Fig. 1). These results indicate that Mfi2 does not associate with Dnm1, suggesting that these proteins functions independently. However, we cannot exclude the possibility of transient or indirect interactions that were not detected under our experimental conditions. Therefore, we have not included these results in the revised manuscript, but we provide this information here in response to the reviewer's comment.

Response Figure 1. No detectable interaction between Mfi2 and Dnm1.

S. cerevisiae cells with (+) or without (-) expression of the indicated proteins (Mfi2-FLAG expressed under the *CUP1* promoter) were cultured in SML until mid-log phase and shifted to SD-N (mitophagy induction). Cell lysates (Input) and anti-FLAG immunoprecipitates (IP: FLAG) were analyzed by immunoblotting using anti-GFP, anti-FLAG, and anti-Mfi2 antibodies. Endogenous Mfi2 was co-immunoprecipitated with Mfi2-FLAG, while Mfi2-Dnm1 interaction was not detected.

3. The authors report a CL-dependent binding of Mfi2 to liposomes. Is the recruitment of Mfi2 to mitochondria impaired when CL-synthesis is blocked, e.g. in *crd1delta* mitochondria?

Response:

We initially attempted to compare the mitochondrial targeting of endogenous Mfi2 in WT and *crd1Δ* cells by subcellular fractionation analysis. However, because *crd1Δ* cells grow poorly and generally show reduced levels of mitochondrial proteins, it was difficult to reliably evaluate mitochondrial targeting of endogenous Mfi2 (Response Fig. 2). We therefore analyzed the localization of GFP-Mfi2 (under the *ADH1* promoter) in WT and *crd1Δ* cells. Although GFP-Mfi2

localized to mitochondria in both cells, cytoplasmic GFP puncta were more frequently observed in *crd1Δ* cells, likely reflecting mistargeted proteins. These data are now included in the revised manuscript (new Fig. 4E), demonstrating that CL deficiency partially impairs efficient mitochondrial targeting of Mfi2. We also performed coarse-grained molecular dynamics (CGMD) simulations of Mfi2-lipid bilayer interaction in the presence or absence of CL. Please see our response to Reviewer #3, major comment 1, for details.

Response Figure 2. Subcellular fractionation analysis of Mfi2 in WT and *crd1Δ* cells.

WT and *crd1Δ* cells were cultured in YPL medium and subjected to subcellular fractionation. Mitochondrial and cytosolic fractions were analyzed by immunoblotting using anti-Mfi2, anti-Cox2 (mitochondria), and anti-Pgk1 (cytosol) antibodies. The reduced mitochondrial protein levels in *crd1Δ* cells made it difficult to reliably evaluate mitochondrial targeting of endogenous Mfi2.

4. Figure 4B: a wild-type control should be added.

Response:

We appreciate Reviewer #1's suggestion to include a WT control in Figure 4B (Fig. 5B in revised version). However, the focus of this figure is on the rescue of mitophagy defects in the *mfi2Δ dnm1Δ* strain, and the key comparison is between this mutant and the rescue conditions. Therefore, we believe that adding a WT control is not essential for the analysis. Statistical analysis has been performed to support our conclusions, and we hope this clarifies our approach.

Reviewer #1 (Significance (Required)):

The reported findings are interesting for a broad readership.

Response:

We appreciate Reviewer #1's recognition of the relevance of our findings to a broad readership.

Reviewer #2 (Evidence, reproducibility and clarity (Required)):

In this study, the authors discover a mitochondrial fission factor, termed Mfi2, that promotes mitophagy efficiency and that functions in a partially redundant way with Dnm1 for the fission of mitochondrial outer membranes during mitophagy. The discovery helps to clarify why Dnm1 does not appear to be essential for fission mediated mitophagy by Dnm1. Mfi2 is structurally similar to the inner membrane fission factor Atg44 which is consistent with Mfi2's fission activity. The authors show that Mfi2 has membrane fission activity towards nanotubes in vitro, and that membrane binding is dependent of high levels of cardiolipin, a mitochondrially enriched lipid. In summary, the authors show that Mfi2 mediates mitochondrial outer membrane fission together with Drp1, whereas Atg44 mediates inner membrane fission, that together are necessary for mitophagy.

Response:

We thank Reviewer #2 for the positive assessment and for clearly summarizing the main contributions of our work.

Major:

1. Figure 2: How do the expression levels of the Mfi2 constructs compare to the endogenous levels of the protein? This will help to gauge to what degree Mfi2 N66 overexpression is needed to achieve mitochondrial fragmentation in Atg44 delta cells and also the low level of mitophagy rescue that was observed.

Response:

We used the *TDH3* promoter for the overexpression of Mfi2 in Figures 2D and 2E. Since our Mfi2 antibody recognizes only the C-terminal region of Mfi2, it can detect full-length Mfi2 but not Mfi2(N66). To address expression levels, we performed immunoblotting using anti-Mfi2 and anti-FLAG antibodies. These analyses showed that full-length Mfi2 was expressed at much higher levels than endogenous Mfi2 (new Fig. EV3A), and that Mfi2-FLAG and Mfi2(N66)-FLAG were expressed at comparable levels (new Fig. EV3B). Therefore, the ability of Mfi2(N66) to induce mitochondrial fragmentation in *atg44Δ* cells and to partially rescue mitophagy is observed only under overexpression conditions. These results have been incorporated into the revised manuscript (Page 7, lines 5-7). Because the expression level of Mfi2-FLAG was lower than that of untagged Mfi2, mitochondrial fragmentation and mitophagy assays were performed using untagged constructs.

2. Figure 3A-B: The cardiolipin binding results in vitro are interesting but the concentration of cardiolipin is much lower on the outer membrane versus the inner membrane. Can the authors comment on whether the cardiolipin levels used on the nanotubes are relevant to that of the mitochondrial outer membrane where Mfi2 is located? Can the authors provide quantitative data for these experiments to help strengthen their conclusions?

Can the authors also use purified MBP alone or a form of Mfi2 that cannot bind to membrane e.g. Mfi2-C33) as a control?

Full Revision

Response:

We thank the reviewer for raising this important point regarding our cardiolipin (CL)-dependent *in vitro* data. In our experiments, we used 20% CL, a concentration higher than the typical levels in the mitochondrial outer membrane, which contains less than 5% CL. However, it is known that CL translocates to the outer membrane under mitophagy-inducing conditions (e.g., Chu et al., *Nat Cell Biol*, 2013; Kagan et al., *Cell Death Differ*, 2016). Thus, one possibility is that the local or transient concentration of CL in the mitochondrial outer membrane may become elevated, and Mfi2 could act at such sites.

Following the reviewer's suggestion and to strengthen our conclusions, we performed a quantitative analysis of the nanotube fission experiments. We measured the percentage of severed tubes under each condition, the total number of tubes analyzed (n), and the relationship between tube diameter and fission efficiency. These data confirmed that Mfi2 and Atg44 exhibit robust fission activity under CL-containing conditions.

Furthermore, we included control experiments using purified MBP alone and the membrane-binding-deficient mutant Mfi2(C33). Both behaved as expected negative controls and did not induce nanotube fission. Together with the quantitative analysis, these results have been incorporated into the revised manuscript as new Figure 3C.

3. Figure 4D: The protrusions are very difficult to visualize. Can the authors also provide zoomed in regions. Is the data representative from 3 or more independent experiments? Can the authors provide a graph of the quantitation to aid readers with analysis of the data?

Response:

We thank the reviewer for this helpful suggestion. In the revised manuscript, we now present the protrusion images at the correct resolution and have also added higher magnification views to improve their visibility (Fig. 5C, D). Importantly, mitochondrial protrusions were consistently observed in the indicated strains but never in core *ATG* deletion mutants, underscoring that their formation is dependent on the autophagy machinery. The data are based on three independent experiments and are also presented as a graph quantifying the frequency of protrusions (Fig. 5D).

4. Figure 4D: It is fascinating to see the mitochondrial protrusion formation being dependent on autophagy factors but not mitochondrial fission factors. To help visualize this, can the authors image one of either Atg1, Atg8 to address whether phagophores are forming on the protrusions and if so where they are positionally located on the protrusion in control and/or *mfi2,dnm1,atg44* triple mutant cells?

Response:

We thank the reviewer for this insightful suggestion. In our previous study (Fukuda *et al.*, *Mol Cell*, 2023), we reported that mitochondrial protrusions are formed in an autophagy-dependent manner, requiring core mitophagy and autophagy factors. Consistently, Atg proteins such as Atg8 were observed to accumulate on these protrusions. In the revised manuscript, we now provide new

imaging data showing GFP-Atg8 puncta on mitochondrial protrusions visualized by Om14-RFP. Importantly, such GFP-Atg8 puncta were also observed on protrusions in the *mfi2Δ dnm1Δ atg44Δ* triple mutant cells (new Fig. 5E). As in our previous analyses, the resolution of our imaging system does not allow precise determination of the positional relationship between phagophores and protrusions; nevertheless, these data confirm that Atg8 puncta are associated with the protrusions themselves. These results have been incorporated into the revised manuscript (Page 9, lines 12-17).

Minor:

1. Is it possible to target Atg44 to the mitochondrial outer membrane, either by attaching an OM anchor or using part of the N-terminus of Mfi2? This will help elucidate how Mfi2 reaches the outer membrane and whether Atg44 can be just as active on the outer membrane as long as it can access it.

Response:

We thank the reviewer for this multifaceted suggestion. We addressed the comment in two ways.

(i) Targeting Atg44 to the outer membrane.

We first constructed two Mfi2-Atg44 chimeras, Mfi2(1-25)-Atg44(28-73) and Mfi2(1-34)-Atg44(38-73), corresponding to fusions starting immediately after the α 1 helix of Mfi2 or after the subsequent loop region, respectively. We also constructed Atg44 variants fused to the C-terminal tail anchors of Tom5 or Fis1 (Atg44-Tom5^{TA} and Atg44-Fis1^{TA}). The Mfi2-Atg44 chimeras were not detected by immunoblot, suggesting instability and preventing functional assessment. Both Atg44-Tom5^{TA} and Atg44-Fis1^{TA} were stably expressed but did not rescue the phenotypes of *mfi2Δ*. Please see Response Figure 3.

Response Figure 3. Attempts to target Atg44 to the mitochondrial outer membrane.

(A) *mfi2Δ dnm1Δ* cells expressing Atg44, Mfi2-Atg44 chimeric proteins, Atg44-Tom5^{TA}, or Atg44-Fis1^{TA} were cultured in SML medium, and cell lysates were subjected to immunoblotting analysis using anti-Atg44 antibody. Expression of Atg44-Tom5^{TA} and Atg44-Fis1^{TA} was confirmed, whereas Mfi2-Atg44 chimeras were not detected. **(B)** *mfi2Δ dnm1Δ* cells expressing Mfi2, Atg44, Atg44-Tom5^{TA}, or Atg44-Fis1^{TA} were cultured in SML medium, shifted to SD-N for 6 h, and cell

lysates were subjected to immunoblotting to assess mitophagy. This result showed that neither Atg44-Tom5^{TA} nor Atg44-Fis1^{TA} rescue Mfi2 function.

(ii) How Mfi2 reaches and functions at the outer membrane.

To further elucidate this point, we performed coarse-grained molecular dynamics (CGMD) simulations, which revealed that Mfi2 remained associated with CL-containing bilayers but not with bilayers lacking CL (new Fig. 4; Movies EV1-EV4). In addition, truncation analysis narrowed down the N-terminal region sufficient for mitochondrial targeting of Mfi2 (new Fig. 2A-C). Deletion of the first α -helix ($\Delta\alpha1$) abolished mitochondrial localization of Mfi2, underscoring its essential role. On the other hand, the $\alpha1$ alone was poorly expressed and could not be reliably assessed (Response Fig. 4). These findings address the reviewer's question and are consistent with our response to Reviewer #3, Major Comment 1, which emphasized the CL-dependence of Mfi2-membrane association. Together, these complementary approaches clarify how Mfi2 reaches the outer membrane and highlight that Atg44 cannot simply substitute for Mfi2 by forced outer-membrane targeting.

Response Figure 4. Expression and localization of the isolated $\alpha1$ helix of Mfi2.

(A) Immunoblot analysis of GFP-Mfi2($\alpha1$) expressed in yeast. The construct showed only faint expression compared with full-length Mfi2. (B) Fluorescence microscopy of GFP-Mfi2($\alpha1$). No detectable GFP signal was observed, consistent with its low expression as shown by immunoblotting. Scale bar, 4 μ m.

2. Are microtubules or actin required for the protrusions to form? Using the triple mutant cells that have a high proportion of protrusions, it could be tried to add cytoskeletal depolymerizing drugs such as nocodazole for microtubules or Latrunculin A or Latrunculin B for actin.

Response:

We thank the reviewer for this suggestion. To address this point, we tested the effect of cytoskeletal depolymerizing drugs on protrusion formation in the *mfi2* Δ *dnm1* Δ *atg44* Δ triple mutant cells. Treatment with nocodazole (for microtubules) or Latrunculin A (for actin) at concentrations confirmed to be effective did not alter the mitochondrial protrusion formation

(Response Fig. 5). These results suggest that protrusion formation during mitophagy does not require intact microtubules or actin filaments.

Response Figure 5. Effect of cytoskeletal depolymerizing drugs on mitochondrial protrusion formation.

(A) WT cells expressing GFP-Tub1 were treated with 20 μ g/ml nocodazole for 30 min, and disruption of cytoplasmic microtubules was confirmed. (B) WT cells expressing Abp1-GFP were treated with 200 μ M Latrunculin A for 30 min, and loss of cortical actin patches was confirmed. (C) *mfi2Δ dnm1Δ atg44Δ* triple mutant cells were cultured in YPL and shifted to SD-N medium for 6 h in the presence or absence of nocodazole or Latrunculin A and analyzed by fluorescence microscopy. Mitochondrial protrusions were observed regardless of drug treatment. Scale bars, 4 μ m.

Reviewer #2 (Significance (Required)):

Significance:

The discovery of Mfi2 as an outer membrane mitophagy fission factor is an exciting, and very important and significant contribution to the field. The data in this study are clear and the conclusions are generally well supported by the experiments. This study appears to be suitable as a report style manuscript given that there is limited mechanistic analysis of Mfi2 activity. This does not affect the importance of the work, it just means that it is suited as a report of a significant discovery. Overall, this fills an important knowledge gap in solving the mystery behind which factors are involved in mitochondrial outer membrane fission during mitophagy, and provides a clarification why Dnm1 loss alone minimally affects mitophagy. This work will appeal to researchers interested in mitochondrial biology, the autophagy field, and cell biologists interested in organelle membrane dynamics, and is also broadly important and interesting to all cell biologists.

Reviewer expertise: mitophagy mechanisms, cell biology of mitophagy, autophagy and autophagosome formation, mitochondrial biology including OXPHOS and mitochondrial dynamics

Full Revision

Response:

We appreciate Reviewer #2's comments on the importance and potential impact of our discovery for the mitophagy and cell biology fields.

Reviewer #3 (Evidence, reproducibility and clarity (Required)):

The manuscript by Furukawa et al. presents a well-structured and thorough study identifying Mfi2 as a novel mitochondrial outer membrane-resident fission factor required for mitophagy in *Saccharomyces cerevisiae*. The authors demonstrate that Mfi2, together with the inner membrane mitofission Atg44 and the dynamin-related GTPase Dnm1, contributes to mitochondrial fragmentation during mitophagy. Importantly, they show that while Dnm1 is dispensable on its own, Mfi2 and Dnm1 act redundantly from the outer membrane to support Atg44-mediated fission. The data are robust, the figures are clear, and the mechanistic insight into how mitophagy-specific fission is achieved is of high relevance to the field of mitochondrial quality control.

Overall, this is a logically constructed and convincing study with important implications for understanding compartment-specific mechanisms of mitochondrial fission during selective autophagy. The conclusions are largely well supported by the data. However, a few issues and points of clarification should be addressed before publication.

Response:

We thank Reviewer #3 for the careful and constructive review and for acknowledging the logical structure and robustness of our data.

Major Comments

1. The observation that both Mfi2 and Atg44 require high cardiolipin (CL) content for membrane binding and fission *in vitro* is intriguing, especially given that CL is enriched in the inner membrane. The authors mention CL externalisation during mitophagy, but this connection could be made more explicit earlier in the manuscript. Furthermore, since the molecular mechanism of membrane interaction remains unresolved, I would strongly encourage the authors to undertake coarse-grained molecular dynamics simulations to explore how Mfi2 might interact with lipid bilayers of differing composition. This could clarify the role of CL and the potential structural contribution of the disordered C-terminal region.

Response:

We thank the reviewer for highlighting the need to clarify the connection between CL externalization and the observed CL-dependent membrane binding and fission activity of Mfi2 and Atg44. While we mentioned CL externalization during mitophagy in the Discussion, we agree that this connection should be made more explicit earlier in the manuscript. In the revised manuscript, we have incorporated a brief rationale in the Results section to clarify that CL translocates to the mitochondrial outer membrane under mitophagy-inducing conditions (page 8, lines 10-13). This provides a physiological basis for our *in vitro* reconstitution assays using CL-containing liposomes.

We also appreciate the reviewer's suggestion to explore the molecular basis of Mfi2-lipid interaction through coarse-grained molecular dynamics (CGMD) simulations. To address this, in collaboration with Dr. Yuji Sakai, we performed CGMD simulations with lipid bilayers of varying composition. In simulations with CL-containing bilayers, full-length Mfi2 was initially placed 3 nm above the bilayer and diffused randomly. Upon contact, Mfi2 associated with the bilayer and remained bound throughout the 1- μ s simulation. In contrast, in the absence of CL, Mfi2 repeatedly

approached the bilayer but did not establish a stable association with it. Similar CL-dependent behavior was observed for the C-terminally truncated mutant Mfi2(N66). These results indicate that CL is critical for stable bilayer association of Mfi2, and that the N-terminal region alone is sufficient for this CL-dependent interaction. These new data are now included in the revised manuscript (new Fig. 4; Movies EV1-EV4).

2. While the genetic and phenotypic data indicate that Mfi2 and Dnm1 act independently to support mitochondrial fission, the spatial and temporal organisation of their activity during mitophagy remains unclear. Do Mfi2 and Dnm1 colocalise at fission sites, or do they act at separate subdomains of the outer membrane? Live-cell imaging with fluorescently tagged Mfi2 and Dnm1, particularly during mitophagy induction, could help clarify whether these factors act in concert or at distinct locations and time points. This would also help determine whether their apparent redundancy reflects parallel mechanisms or functional compensation at shared sites. It would also be interesting to combine this with Atg44.

Response:

We thank the reviewer for this insightful comment. To address whether Mfi2 and Dnm1 localize to the same fission sites during mitophagy, we analyzed cells undergoing mitophagy induction. In *atg44Δ* cells, where mitochondrial protrusions accumulate, Dnm1-GFP puncta were detected at approximately 60% of the protrusions, indicating that Dnm1 accumulates at these sites (Response Fig. 6A). In contrast, GFP-Mfi2 was distributed uniformly along mitochondria and did not form puncta (Response Fig. 6B), which may reflect a limitation of the GFP fusion protein. Indeed, Om45-RFP processing assays showed that cells expressing GFP-Mfi2 have reduced mitophagy activity compared with WT cells (untagged Mfi2), suggesting partial loss of function of Mfi2 by GFP tagging (Response Fig. 6C). It is also challenging to analyze Mfi2, Dnm1, Atg44, and mitochondrial fission sites simultaneously, as fluorescence-tagged Atg44 has been shown to lose its function (Fukuda *et al.*, *Mol Cell*, 2023).

Thus, while our genetic and phenotypic data support independent roles of Mfi2 and Dnm1, elucidating their precise spatial and temporal organization including Atg44 will require improved tagging strategies and is an important goal for future work.

Response Figure 6. Localization of Dnm1-GFP and GFP-Mfi2 during mitophagy induction.

(A) *atg44Δ* cells expressing Dnm1-GFP were cultured in YPL until mid-log phase and shifted to SD-N for 6 h to induce mitophagy. Dnm1-GFP puncta were detected at approximately half of the mitochondrial protrusions (arrows). Scale bar, 2 μm. (B) *atg44Δ* cells expressing GFP-Mfi2 under the control of the *ADH1* (strong) or *TDH3* (very strong) promoter were cultured in YPL until mid-log phase and shifted to SD-N for 6 h to induce mitophagy. GFP-Mfi2 was distributed uniformly along mitochondria and did not form puncta around mitochondrial protrusions (arrows). Scale bar, 2 μm. (C) Om45-RFP processing assay to assess mitophagy activity. WT, *mfi2Δ*, and GFP-Mfi2 (*ADH1* promoter) cells were cultured in YPL until mid-log phase and shifted to SD-N for 6 h. Mitophagy was reduced in both *mfi2Δ* and GFP-Mfi2 cells compared with WT cells, suggesting that GFP tagging compromises Mfi2 function.

Minor Comments

1. The sodium carbonate extraction and proteinase K assays (Figure 1E-F) are standard but may not be familiar to all readers. A brief explanatory sentence clarifying what these methods reveal about membrane topology would improve accessibility.

Response:

We thank the reviewer for this helpful comment. We have added a brief explanatory sentence in the revised manuscript to clarify the principles and interpretation of the sodium carbonate extraction and proteinase K assays (Page 5, lines 25-27; Page 6, lines 3-5).

2. While immunoblot quantifications are shown throughout, it would be helpful to include statistical analysis where appropriate, especially in cases where differences between genotypes or constructs are modest.

Response:

Statistical analyses have been added for immunoblot quantifications where appropriate, particularly in cases where differences between genotypes or constructs are modest.

3. The naming of Mfi2 as a mitofissin is consistent with previous terminology introduced for Atg44, but the term remains relatively new. A brief clarification distinguishing "mitofissin" from the better-known "mitofusin" family in mammals would help avoid confusion for readers less familiar with yeast-specific nomenclature.

Response:

We have added a brief explanation of the term "mitofissin" to distinguish it from the mammalian "mitofusin" family in Introduction (Page 3, line 26-Page 4 line 1).

Reviewer #3 (Significance (Required)):

This is a strong and well-executed study that provides mechanistic insight into how mitochondrial fission is coordinated during mitophagy in yeast. A major strength is the identification and characterisation of Mfi2 as a previously unrecognised outer membrane fission factor acting in parallel with Dnm1 and in coordination with the intermembrane space protein Atg44. The genetic,

Full Revision

imaging, and in vitro biochemical data are carefully integrated, and the authors are transparent about limitations, including open questions around the C-terminal domain of Mfi2, CL dependence, and the evolutionary conservation of mitofissins.

The work makes a conceptual advance by showing that mitophagy-specific mitochondrial fission requires the cooperation of spatially separated factors acting from both the inside and outside of mitochondria, a mechanism that had not been fully appreciated. This study helps resolve previous contradictions regarding the dispensability of Dnm1 in mitophagy, thereby filling a gap in our understanding of organelle-specific fission. While the findings are focused on yeast, they raise broader questions about whether similar principles apply to higher eukaryotes (historically yeast research was always at the forefront of autophagy field).

The study will be of interest to specialists in autophagy, mitochondrial dynamics, and yeast cell biology, as well as researchers working on membrane remodelling and organelle quality control. While the audience is primarily specialised, the conceptual insights will resonate more broadly in the cell biology community.

I am an expert in mitophagy mechanisms in mammalian cells, and while not a specialist in yeast models, I found the study logical, rigorous, and of clear relevance to the broader autophagy field.

Response:

We are grateful for Reviewer #3's recognition of the conceptual advance provided by our study and its relevance beyond yeast biology.

Dear Prof. Kanki,

Thank you for the submission of your revised manuscript to EMBO reports. We have now received the full set of referee reports that is copied below. As you will see, all referees are very positive about the study and recommend publication.

Browsing through the manuscript myself, I noticed a few editorial things that we need before we can proceed with the official acceptance of your study.

- 1) Please reduce the number of keywords to 5.
- 2) Please place the Disclosure and competing interests statement after the Acknowledgments.
- 3) Regarding the Author Contributions, we now use CRedit to specify the contributions of each author in the journal submission system. Therefore, please remove the Author Contributions from the manuscript file and make sure that the author contributions in our online manuscript tracking system are correct and up-to-date. The information you specified in the system will be automatically retrieved and typeset into the article. You can enter additional information in the free text box provided, if you wish. See also our guide to authors <https://www.embopress.org/page/journal/14693178/authorguide#authorshipguidelines>.
- 4) You refer to "our unpublished data" on p12. Please note our relevant editorial policy that does not allow to base statements on "data not shown". Please either provide the relevant data or remove the conclusion/statement from the manuscript.
- 5) Table EV1 and Table EV2 should be removed from the manuscript and uploaded as separate Expanded View Content files (the legends should also be part of the table files, not in the manuscript).
- 6) The Author Checklist refers to Table EV1 for the information on microbes (line 63). I think this should be Table EV2.
- 7) Please remove the movie legends from the manuscript and provide them in separate text files; then each movie should be zipped up with its corresponding legend so that we have 4 zip folders separately uploaded (Movie EV1-Movie EV4).
- 8) During our routine image integrity checks, we observed that the blot images within the figure set and in the corresponding source data appear pixelated under analysis. The background information is largely absent, and the blots appear over-contrasted. This is often a result of converting original 16-bit TIFF files to RGB format for publication. While this is not inherently problematic, it can give the impression of image alteration to critical readers.

To address this, please upload the blot figures at their original captured resolution. Please also upload the original blot source data using the original captured image with your online submission or by depositing the raw files on BioStudies <https://www.ebi.ac.uk/biostudies/sourcedata/studies> and including the archive accession number in your Data Availability section.

This will enable us to confirm the integrity of the complete figure set and enhance transparency for readers.

- 9) Source data: Please provide the Western blots for Figure 1 as independent files and sort them into separate folders per figure panel. The same holds true for Figure 2B and 2D, Figure 5A and 5B, and EV3A and EV3B. We need one subfolder per figure panel in all cases. It would be good to add the molecular weight marker images for these gels.

- 10) Please address the following comments in the figure legends:
 - Please indicate the statistical test used for data analysis in the legend of figure 4E
 - Please note that the error bars are not defined in the legend of figure 4E
 - Please note that the asterisk is not defined in the legend of figure EV3 A. This needs to be rectified.

With kind regards,

=====

Referee #1:

The authors addressed most of my concerns. I recommend publication of this article.

Referee #2:

The authors adequately responded to my comments and the manuscript can now be recommended for publication.

Referee #3:

The authors have done an excellent job addressing all the comments. Congratulations on a great piece of work.

Referee #1:

The authors addressed most of my concerns. I recommend publication of this article.

Referee #2:

The authors adequately responded to my comments and the manuscript can now be recommended for publication.

Referee #3:

The authors have done an excellent job addressing all the comments. Congratulations on a great piece of work.

Rev_Com_number: RC-2025-03004

New_manu_number: EMBOR-2025-62137V2

Corr_author: Kanki

Title: Mitochondrial fission during mitophagy requires both inner and outer mitofissins

All minor editorial requests have been addressed by the authors.

Prof. Tomotake Kanki
Kyushu University Graduate School of Medical Sciences
Department of Cellular Physiology
951-8510
Japan

Dear Prof. Kanki,

I am very pleased to accept your manuscript for publication in the next available issue of EMBO reports. Thank you for your contribution to our journal.

You may qualify for financial assistance for your publication charges - either via a Springer Nature fully open access agreement or an EMBO initiative. Check your eligibility: <https://link.springer.com/journal/44319/how-to-publish-with-us>

Yours sincerely,

>>> Please note that it is EMBO Reports policy for the transcript of the editorial process (containing referee reports and your response letter) to be published as an online supplement to each paper. If you do NOT want this, you will need to inform the Editorial Office via email immediately. More information is available here: <https://link.springer.com/partners/embo-press/editorial-policies#Peer%20review>

Rev_Com_number: RC-2025-03004
New_manu_number: EMBOR-2025-62137V3
Corr_author: Kanki
Title: Mitochondrial fission during mitophagy requires both inner and outer mitofissins